# ROBUSTNESS VERIFICATION FOR TRANSFORMERS

**Zhouxing Shi**[1], **Huan Zhang**[2], **Kai-Wei Chang**[2], **Minlie Huang**[1], **Cho-Jui Hsieh**[2]
[1]Dept. of Computer Science & Technology, Tsinghua University, Beijing 10084, China
[2]Dept. of Computer Science, University of California, Los Angeles, CA 90095, USA
`zhouxingshichn@gmail.com, huan@huan-zhang.com`
`kw@kwchang.net, aihuang@tsinghua.edu.cn, chohsieh@cs.ucla.edu`

## ABSTRACT

Robustness verification that aims to formally certify the prediction behavior of neural networks has become an important tool for understanding model behavior and obtaining safety guarantees. However, previous methods can usually only handle neural networks with relatively simple architectures. In this paper, we consider the robustness verification problem for Transformers. Transformers have complex self-attention layers that pose many challenges for verification, including cross-nonlinearity and cross-position dependency, which have not been discussed in previous works. We resolve these challenges and develop the first robustness verification algorithm for Transformers. The certified robustness bounds computed by our method are significantly tighter than those by naive Interval Bound Propagation. These bounds also shed light on interpreting Transformers as they consistently reflect the importance of different words in sentiment analysis.

## 1 INTRODUCTION

Deep neural networks have been successfully applied to many domains. However, these black-box models are generally difficult to analyze and their behavior is not guaranteed. Moreover, it has been shown that the predictions of deep networks become unreliable and unstable when tested in unseen situations, e.g., in the presence of small adversarial perturbations to the input (Szegedy et al., 2013; Goodfellow et al., 2014; Lin et al., 2019). Therefore, neural network verification has become an important tool for analyzing and understanding the behavior of neural networks, with applications in safety-critical applications (Katz et al., 2017; Julian et al., 2019; Lin et al., 2019), model explanation (Shih et al., 2018) and robustness analysis (Tjeng et al., 2019; Wang et al., 2018c; Gehr et al., 2018; Wong & Kolter, 2018; Singh et al., 2018; Weng et al., 2018; Zhang et al., 2018).

Formally, a neural network verification algorithm aims to provably characterize the prediction of a network within some input space. For example, given a $K$-way classification model $f : \mathbb{R}^d \to \mathbb{R}^K$, where $f_i(\mathbf{x})$ stands for the predicted score of class $i$, we can verify some linear specification (defined by a vector $\mathbf{c}$) as below:

$$\min_{\mathbf{x}} \sum_i \mathbf{c}_i f_i(\mathbf{x}) \quad \text{s.t. } \mathbf{x} \in \mathbb{S}, \tag{1}$$

where $\mathbb{S}$ is a predefined input space. In the **robustness verification problem**, $\mathbb{S} = \{\mathbf{x} \mid \|\mathbf{x} - \mathbf{x}_0\|_p \leq \epsilon\}$ is defined as some small $\ell_p$-ball around the original example $\mathbf{x}_0$, and setting up $\mathbf{c} = 1_{y_0} - 1_y$ enables us to verify whether the logit output of class $y_0$ is always greater than another class $y$ for any input within $\mathbb{S}$. This is a nonconvex optimization problem which makes computing the exact solution challenging, and thus several algorithms are recently proposed to find the lower bounds of Eq. (1) in order to efficiently obtain a safety guarantee (Gehr et al., 2018; Weng et al., 2018; Zhang et al., 2018; Singh et al., 2019). Moreover, extensions of these algorithms can be used for verifying properties beyond robustness, such as rotation or shift invariant (Singh et al., 2019), conservation of energy (Qin et al., 2019) and model correctness (Yang & Rinard, 2019).

However, most of existing verification methods focus on relatively simple neural network architectures, such as feed-forward and recurrent neural networks, while they cannot handle complex structures. In this paper, we develop the first robustness verification algorithm for Transformers (Vaswani et al., 2017) with self-attention layers. Transformers have been widely used in natural language processing (Devlin et al., 2019; Yang et al., 2019; Liu et al., 2019) and many other domains (Parmar

et al., 2018; Kang & McAuley, 2018; Li et al., 2019b; Su et al., 2019; Li et al., 2019a). For frames under perturbation in the input sequence, we aim to compute a lower bound $\epsilon$ such that when these frames are perturbed within $\ell_p$-balls centered at the original frames respectively and with a radius of $\epsilon$, the model prediction is certified to be unchanged. To compute such bounds efficiently, we adopt the linear-relaxation framework (Weng et al., 2018; Zhang et al., 2018) – we recursively propagate and compute linear lower and upper bounds for each neuron w.r.t the input within the perturbation space $S$.

We resolve several particular challenges in verifying Transformers. **First**, Transformers with self-attention layers have a complicated architecture. Unlike simpler networks, they cannot be written as multiple layers of affine transformations or element-wise activation functions. Therefore, we need to propagate linear bounds differently for self-attention layers. **Second**, dot products, softmax, and weighted summation in self-attention layers involve multiplication or division of two variables both under perturbation, namely cross-nonlinearity, which is not present in feed-forward networks. Ko et al. (2019) proposed a gradient descent based approach to find linear bounds, however it is inefficient and poses a computational challenge for Transformer verification since self-attention is the core of Transformers. In contrast, we derive closed-form linear bounds that can be computed in $O(1)$ complexity. **Third**, in the computation of self-attention, output neurons in each position depend on *all* input neurons from different positions (namely *cross-position dependency*), unlike the case in recurrent neural networks where outputs depend on only the hidden features from the previous position and the current input. Previous works (Zhang et al., 2018; Weng et al., 2018; Ko et al., 2019) have to track all such dependency and thus is costly in time and memory. To tackle this, we introduce an efficient bound propagating process in a forward manner specially for self-attention layers, enabling the tighter backward bounding process for other layers to utilize bounds computed by the forward process. In this way, we avoid cross-position dependency in the backward process which is relatively slower but produces tighter bounds. Combined with the forward process, the complexity of the backward process is reduced by $O(n)$ for input length $n$, while the computed bounds remain comparably tight. Our contributions are summarized below:

- We propose an effective and efficient algorithm for verifying the robustness of Transformers with self-attention layers. To our best knowledge, this is the first method for verifying Transformers.
- We resolve key challenges in verifying Transformers, including cross-nonlinearity and cross-position dependency. Our bounds are significantly tighter than those by adapting Interval Bound Propagation (IBP) (Mirman et al., 2018; Gowal et al., 2018).
- We quantitatively and qualitatively show that the certified bounds computed by our algorithm consistently reflect the importance of input words in sentiment analysis, which justifies that these bounds are meaningful in practice and they shed light on interpreting Transformers.

## 2 RELATED WORK

**Robustness Verification for Neural Networks.** Given an input $\mathbf{x}_0$ and a small region $\mathbb{B}_p(\mathbf{x}_0, \epsilon) := \{\mathbf{x} \mid \|\mathbf{x} - \mathbf{x}_0\|_p \leq \epsilon\}$, the goal of robustness verification is to verify whether the prediction of the neural network is unchanged within this region. This problem can be mathematically formulated as Eq. (1). If Eq. (1) can be solved optimally, then we can derive the minimum adversarial perturbation of $\mathbf{x}$ by conducting binary search on $\epsilon$. Equivalently, we obtain the maximum $\epsilon$ such that any perturbation within $\mathbb{B}_p(\mathbf{x}_0, \epsilon)$ cannot change the predicted label.

Several works focus on solving Eq. (1) exactly and optimally, using mixed integer linear programming (MILP) (Tjeng et al., 2019; Dutta et al., 2018), branch and bound (BaB) (Bunel et al., 2018), and satisfiability modulo theory (SMT) (Ehlers, 2017; Katz et al., 2017). Unfortunately, due to the nonconvexity of model $f$, solving Eq. (1) is NP-hard even for a simple ReLU network (Katz et al., 2017). Therefore, we can only expect to compute a lower bound of Eq. (1) efficiently by using relaxations. Many algorithms can be seen as using convex relaxations for non-linear activation functions (Salman et al., 2019), including using duality (Wong & Kolter, 2018; Dvijotham et al., 2018), abstract domains (Gehr et al., 2018; Singh et al., 2018; Mirman et al., 2018; Singh et al., 2019), layer-by-layer reachability analysis (Wang et al., 2018b; Weng et al., 2018; Zhang et al., 2018; Gowal et al., 2018) and semi-definite relaxations (Raghunathan et al., 2018; Dvijotham et al., 2019).

Additionally, robustness verification can rely on analysis on local Lipschitz constants (Hein & Andriushchenko, 2017; Zhang et al., 2019). However, existing methods are mostly limited to verifying networks with relatively simple architectures, such as feed-forward networks and RNNs (Wang et al., 2018a; Akintunde et al., 2019; Ko et al., 2019), while none of them are able to handle Transformers.

**Transformers and Self-Attentive Models.** Transformers (Vaswani et al., 2017) based on self-attention mechanism, further with pre-training on large-scale corpora, such as BERT (Devlin et al., 2019), XLNet (Yang et al., 2019), RoBERTa (Liu et al., 2019), achieved state-of-the-art performance on many NLP tasks. Self-attentive models are also useful beyond NLP, including VisualBERT on vision and language applications (Li et al., 2019b; Su et al., 2019), image transformer for image generation (Parmar et al., 2018), acoustic models for speech recognition (Zhou et al., 2018), sequential recommendation (Kang & McAuley, 2018) and graph embedding (Li et al., 2019a).

The robustness of NLP models has been studied, especially many methods have been proposed to generate adversarial examples (Papernot et al., 2016; Jia & Liang, 2017; Zhao et al., 2017; Alzantot et al., 2018; Cheng et al., 2018; Ebrahimi et al., 2018; Shi et al., 2019). In particular, Hsieh et al. (2019) showed that Transformers are more robust than LSTMs. However, there is not much work on robustness verification for NLP models. Ko et al. (2019) verified RNN/LSTM. Jia et al. (2019); Huang et al. (2019) used Interval Bound Propagation (IBP) for certified robustness training of CNN and LSTM. In this paper, we propose the first verification method for Transformers.

## 3 METHODOLOGY

We aim to verify the robustness of a Transformer whose input is a sequence of frames $\mathbf{X} = [\mathbf{x}^{(1)}, \mathbf{x}^{(2)}, \cdots, \mathbf{x}^{(n)}]$. We take binary text classification as a running example, where $\mathbf{x}^{(i)}$ is a word embedding and the model outputs a score $y_c(\mathbf{X})$ for each class $c$ ($c \in \{0, 1\}$). Nevertheless, our method for verifying Transformers is general and can also be applied in other applications.

For a clean input sequence $\mathbf{X}_0 = [\mathbf{x}_0^{(1)}, \mathbf{x}_0^{(2)}, \cdots, \mathbf{x}_0^{(n)}]$ correctly classified by the model, let $P = \{r_1, r_2, \cdots, r_t\}(1 \leq r_k \leq n)$ be the set of perturbed positions, where $t$ is the number of perturbed positions. Thus the perturbed input belongs to $\mathbb{S}_\epsilon := \{\mathbf{X} = [\mathbf{x}^{(1)}, \mathbf{x}^{(2)}, \cdots, \mathbf{x}^{(n)}] : \|\mathbf{x}^{(r_k)} - \mathbf{x}_0^{(r_k)}\|_p \leq \epsilon, 1 \leq k \leq t, \mathbf{x}^{(i)} = \mathbf{x}_0^{(i)}, \forall i \notin P\}$. Assuming that $c$ is the gold class, the goal of robustness verification is to compute

$$\left\{ \min_{\mathbf{X} \in \mathbb{S}} y_c(\mathbf{X}) - y_{1-c}(\mathbf{X}) \right\} := \delta_\epsilon.$$

If $\delta_\epsilon > 0$, the output score of the correct class is always larger than the incorrect one for any input within $\mathbb{S}_\epsilon$. As mentioned previously, computing the exact values of $\delta_\epsilon$ is NP-hard, and thus our goal is to efficiently compute a lower bound $\delta_\epsilon^L \leq \delta_\epsilon$.

### 3.1 BASE FRAMEWORK

We obtain $\delta_\epsilon^L(\mathbf{X})$ by computing the bounds of each neuron when $\mathbf{X}$ is perturbed within $\mathbb{S}_\epsilon$ ($\delta_\epsilon^L$ can be regarded as a final neuron). A Transformer layer can be decomposed into a number of sub-layers, where each sub-layer contains neurons after some operations. These operations can be categorized into three categories: 1) linear transformations, 2) unary nonlinear functions, and 3) operations in self-attention. Each sub-layer contains $n$ positions in the sequence and each position contains a group of neurons. We assume that the Transformer we verify has $m$ sub-layers in total, and the value of the $j$-th neuron at the $i$-th position in the $l$-th sub-layer is $\Phi_j^{(l,i)}(\mathbf{X})$, where $\Phi^{(l,i)}(\mathbf{X})$ is a vector for the specified sub-layer and position. Specially, $\Phi^{(0,i)} = \mathbf{x}^{(i)}$ taking $l = 0$. We aim to compute a global lower bound $f_j^{(l,i),L}$ and a global upper bound $f_j^{(l,i),U}$ of $\Phi_j^{(l,i)}(\mathbf{X})$ for $\mathbf{X} \in \mathbb{S}_\epsilon$.

We compute bounds from the first sub-layer to the last sub-layer. For neurons in the $l$-th layer, we aim to represent their bounds as linear functions of neurons in a previous layer, the $l'$-th layer:

$$\sum_{k=1}^n \mathbf{\Lambda}_{j,:}^{(l,i,l',k),L} \Phi^{(l',k)}(\mathbf{X}) + \mathbf{\Delta}_j^{(l,i,l'),L} \leq \Phi_j^{(l,i)}(\mathbf{X}) \leq \sum_{k=1}^n \mathbf{\Lambda}_{j,:}^{(l,i,l',k),U} \Phi^{(l',k)}(\mathbf{X}) + \mathbf{\Delta}_j^{(l,i,l'),U}, \quad (2)$$

where $\mathbf{\Lambda}^{(l,i,l',k),L}, \mathbf{\Delta}^{(l,i,l'),L}$ and $\mathbf{\Lambda}^{(l,i,l',k),U}, \mathbf{\Delta}^{(l,i,l'),U}$ are parameters of linear lower and upper bounds respectively. Using linear bounds enables us to efficiently compute bounds with a reasonable tightness. We initially have $\mathbf{\Lambda}^{(l,i,l,i),L} = \mathbf{\Lambda}^{(l,i,l,i),U} = \mathbf{I}$ and $\mathbf{\Delta}^{(l,i,l),L} = \mathbf{\Delta}^{(l,i,l),U} = \mathbf{0}$. Thereby the right-hand-side of Eq. (2) equals to $\Phi_j^{(l,i)}(\mathbf{X})$ when $l' = l$. Generally, we use a backward process to propagate the bounds to previous sub-layers, by substituting $\Phi^{(l',i)}$ with linear functions of previous neurons. It can be recursively conducted until the input layer $l' = 0$. Since $\Phi^{(0,k)} = \mathbf{x}^{(k)} = \mathbf{x}_0^{(k)} (\forall k \notin P)$ is constant, we can regard the bounds as linear functions of the perturbed embeddings $\Phi^{(0,r_k)} = \mathbf{x}^{(r_k)} (1 \le k \le t)$, and take the global bounds for $\mathbf{x}^{(r_k)} \in \mathbb{B}_p(\mathbf{x}_0^{(r_k)}, \epsilon)$:

$$f_j^{(l,i),L} = -\epsilon \sum_{k=1}^{t} \| \mathbf{\Lambda}_{j,:}^{(l,i,0,r_k),L} \|_q + \sum_{k=1}^{n} \mathbf{\Lambda}_{j,:}^{(l,i,0,k),L} \mathbf{x}_0^{(k)} + \mathbf{\Delta}_j^{(l,i,0),L}, \tag{3}$$

$$f_j^{(l,i),U} = \epsilon \sum_{k=1}^{t} \| \mathbf{\Lambda}_{j,:}^{(l,i,0,r_k),U} \|_q + \sum_{k=1}^{n} \mathbf{\Lambda}_{j,:}^{(l,i,0,k),U} \mathbf{x}_0^{(k)} + \mathbf{\Delta}_j^{(l,i,0),U}, \tag{4}$$

where $1/p + 1/q = 1$ with $p, q \ge 1$. These steps resemble to CROWN (Zhang et al., 2018) which is proposed to verify feed-forward networks. We further support verifying self-attentive Transformers which are more complex than feed-forward networks. Moreover, unlike CROWN that conducts a fully backward process, we combine the backward process with a forward process (see Sec. 3.3) to reduce the computational complexity of verifying Transformers.

## 3.2 LINEAR TRANSFORMATIONS AND UNARY NONLINEAR FUNCTIONS

Linear transformations and unary nonlinear functions are basic operations in neural networks. We show how bounds Eq. (2) at the $l'$-th sub-layer are propagated to the $(l' - 1)$-th layer.

**Linear Transformations**  If the $l'$-th sub-layer is connected with the $(l' - 1)$-th sub-layer with a linear transformation $\Phi^{(l',k)}(\mathbf{X}) = \mathbf{W}^{(l')}\Phi^{(l'-1,k)}(\mathbf{X}) + \mathbf{b}^{(l')}$ where $\mathbf{W}^{(l')}, \mathbf{b}^{(l')}$ are parameters of the linear transformation, we propagate the bounds to the $(l'-1)$-th layer by substituting $\Phi^{(l',k)}(\mathbf{X})$:

$$\mathbf{\Lambda}^{(l,i,l'-1,k),L/U} = \mathbf{\Lambda}^{(l,i,l',k),L/U} \mathbf{W}^{(l')}, \mathbf{\Delta}^{(l,i,l'-1),L/U} = \mathbf{\Delta}^{(l,i,l'),L/U} + \left( \sum_{k=1}^{n} \mathbf{\Lambda}^{(l,i,l',k),L/U} \right) \mathbf{b}^{(l')},$$

where "$L/U$" means that the equations hold for both lower bounds and upper bounds respectively.

**Unary Nonlinear Functions**  If the $l'$-th layer is obtained from the $(l' - 1)$-th layer with an unary nonlinear function $\Phi_j^{(l',k)}(\mathbf{X}) = \sigma^{(l')}(\Phi_j^{(l'-1,k)}(\mathbf{X}))$, to propagate linear bounds over the nonlinear function, we first bound $\sigma^{(l')}(\Phi_j^{(l'-1,k)}(\mathbf{X}))$ with two linear functions of $\Phi_j^{(l'-1,k)}(\mathbf{X})$:

$$\alpha_j^{(l',k),L}\Phi_j^{(l'-1,k)}(\mathbf{X}) + \beta_j^{(l',k),L} \le \sigma^{(l')}(\Phi_j^{(l'-1,k)}(\mathbf{X})) \le \alpha_j^{(l',k),U}\Phi_j^{(l'-1,k)}(\mathbf{X}) + \beta_j^{(l',k),U},$$

where $\alpha_j^{(l',k),L/U}, \beta_j^{(l',k),L/U}$ are parameters such that the inequation holds true for all $\Phi_j^{(l'-1,k)}(\mathbf{X})$ within its bounds computed previously. Such linear relaxations can be done for different functions, respectively. We provide detailed bounds for functions involved in Transformers in Appendix B.

We then back propagate the bounds:

$$\mathbf{\Lambda}_{:,j}^{(l,i,l'-1,k),L/U} = \alpha_j^{(l',k),L/U}\mathbf{\Lambda}_{:,j,+}^{(l,i,l',k),L/U} + \alpha_j^{(l',k),U/L}\mathbf{\Lambda}_{:,j,-}^{(l,i,l',k),L/U},$$

$$\mathbf{\Delta}_j^{(l,i,l'-1),L/U} = \mathbf{\Delta}_j^{(l,i,l')，L/U} + \left( \sum_{k=1}^{n} \beta_j^{(l',k),L/U}\mathbf{\Lambda}_{:,j,+}^{(l,i,l',k),L/U} + \beta_j^{(l',k),U/L}\mathbf{\Lambda}_{:,j,-}^{(l,i,l',k),L/U} \right),$$

where $\mathbf{\Lambda}_{:,j,+}^{(l,i,l',k),L/U}$ and $\mathbf{\Lambda}_{:,j,-}^{(l,i,l',k),L/U}$ mean to retain positive and negative elements in vector $\mathbf{\Lambda}_{:,j}^{(l,i,l',k),L/U}$ respectively and set other elements to 0.

### 3.3 SELF-ATTENTION MECHANISM

Self-attention layers are the most challenging parts for verifying Transformers. We assume that $\Phi^{(l-1,i)}(\mathbf{X})$ is the input to a self-attention layer. We describe our method for computing bounds for one attention head, and bounds for different heads of the multi-head attention in Transformers can be easily concatenated. $\Phi^{(l-1,i)}(\mathbf{X})$ is first linearly projected to queries $\mathbf{q}^{(l,i)}(\mathbf{X})$, keys $\mathbf{k}^{(l,i)}(\mathbf{X})$, and values $\mathbf{v}^{(l,i)}(\mathbf{X})$ with different linear projections, and their bounds can be obtained as described in Sec. 3.2. We also keep their linear bounds that are linear functions of the perturbed embeddings. For convenience, let $\mathbf{x}^{(r)} = \mathbf{x}^{(r_1)} \oplus \mathbf{x}^{(r_2)} \oplus \cdots \mathbf{x}^{(r_t)}$, where $\oplus$ indicates vector concatenation, and thereby we represent the linear bounds as linear functions of $\mathbf{x}^{(r)}$:

$$\mathbf{\Omega}_{j,:}^{(l,i),q/k/v,L}\mathbf{x}^{(r)} + \mathbf{\Theta}_j^{(l,i),q/k/v,L} \leq (\mathbf{q/k/v})_j^{(l,i)}(\mathbf{X}) \leq \mathbf{\Omega}_{j,:}^{(l,i),q/k/v,U}\mathbf{x}^{(r)} + \mathbf{\Theta}_j^{(l,i),q/k/v,U},$$

where $q/k/v$ and $\mathbf{q}/\mathbf{k}/\mathbf{v}$ mean that the inequation holds true for queries, keys and values respectively. We then bound the output of the self-attention layer starting from $\mathbf{q}^{(l,i)}(\mathbf{X})$, $\mathbf{k}^{(l,i)}(\mathbf{X})$, $\mathbf{v}^{(l,i)}(\mathbf{X})$.

**Bounds of Multiplications and Divisions** We bound multiplications and divisions in the self-attention mechanism with linear functions. We aim to bound bivariate function $z = xy$ or $z = \frac{x}{y}(y > 0)$ with two linear functions $z^L = \alpha^L x + \beta^L y + \gamma^L$ and $z^U = \alpha^U x + \beta^U y + \gamma^U$, where $x \in [l_x, u_x], y \in [l_y, u_y]$ are bounds of $x, y$ obtained previously. For $z = xy$, we derive optimal parameters: $\alpha^L = l_y, \alpha^U = u_y, \beta^L = \beta^U = l_x, \gamma^L = -l_x l_y, \gamma^U = -l_x u_y$. We provide a proof in Appendix C. However, directly bounding $z = \frac{x}{y}$ is tricky; fortunately, we can bound it indirectly by first bounding a unary function $\overline{y} = \frac{1}{y}$ and then bounding the multiplication $z = x\overline{y}$.

**A Forward Process** For the self-attention mechanism, instead of using the backward process like CROWN (Zhang et al., 2018), we compute bounds with a forward process which we will show later that it can reduce the computational complexity. Attention scores are computed from $\mathbf{q}^{(l,i)}(\mathbf{X})$ and $\mathbf{k}^{(l,i)}(\mathbf{X})$: $\mathbf{S}_{i,j}^{(l)} = (\mathbf{q}^{(l,i)}(\mathbf{X}))^T\mathbf{k}^{(l,j)}(\mathbf{X}) = \sum_{k=1}^{d_{qk}} \mathbf{q}_k^{(l,i)}(\mathbf{X})\mathbf{k}_k^{(l,j)}(\mathbf{X})$, where $d_{qk}$ is the dimension of $\mathbf{q}^{(l,i)}(\mathbf{X})$ and $\mathbf{k}^{(l,j)}(\mathbf{X})$. For each multiplication $\mathbf{q}_k^{(l,i)}(\mathbf{X})\mathbf{k}_k^{(l,j)}(\mathbf{X})$, it is bounded by:

$$\mathbf{q}_k^{(l,i)}(\mathbf{X})\mathbf{k}_k^{(l,j)}(\mathbf{X}) \geq \alpha_k^{(l,i,j),L}\mathbf{q}_k^{(l,i)}(\mathbf{X}) + \beta_k^{(l,i,j),L}\mathbf{k}_k^{(l,j)}(\mathbf{X}) + \gamma_k^{(l,i,j),L},$$

$$\mathbf{q}_k^{(l,i)}(\mathbf{X})\mathbf{k}_k^{(l,j)}(\mathbf{X}) \leq \alpha_k^{(l,i,j),U}\mathbf{q}_k^{(l,i)}(\mathbf{X}) + \beta_k^{(l,i,j),U}\mathbf{k}_k^{(l,j)}(\mathbf{X}) + \gamma_k^{(l,i,j),U}.$$

We then obtain the bounds of $\mathbf{S}_{i,j}^{(l)}$:

$$\mathbf{\Omega}_{j,:}^{(l,i),s,L}\mathbf{x}^{(r)} + \mathbf{\Theta}_j^{(l,i),s,L} \leq \mathbf{S}_{i,j}^{(l)} \leq \mathbf{\Omega}_{j,:}^{(l,i),s,U}\mathbf{x}^{(r)} + \mathbf{\Theta}_j^{(l,i),s,U},$$

$$\mathbf{\Omega}_{j,:}^{(l,i),s,L/U} = \alpha_k^{(l,i,j),L/U}\Big(\sum_{\alpha_k^{(l,i,j),L/U}>0} \mathbf{\Omega}_{k,:}^{(l,i),q,L/U} + \sum_{\alpha_k^{(l,i,j),L/U}<0} \mathbf{\Omega}_{k,:}^{(l,i),q,U/L}\Big)+$$
$$\beta_k^{(l,i,j),L/U}\Big(\sum_{\beta_k^{(l,i,j),L/U}>0} \mathbf{\Omega}_{k,:}^{(l,j),k,L/U} + \sum_{\beta_k^{(l,i,j),L/U}<0} \mathbf{\Omega}_{k,:}^{(l,j),k,U/L}\Big),$$
$$\mathbf{\Theta}_j^{(l,i),s,L/U} = \alpha_k^{(l,i,j),L/U}\Big(\sum_{\alpha_k^{(l,i,j),L/U}>0} \mathbf{\Theta}_k^{(l,i),q,L/U} + \sum_{\alpha_k^{(l,i,j),L/U}<0} \mathbf{\Theta}_k^{(l,i),q,U/L}\Big)+$$
$$\beta_k^{(l,i,j),L/U}\Big(\sum_{\beta_k^{(l,i,j),L/U}>0} \mathbf{\Theta}_k^{(l,j),k,L/U} + \sum_{\beta_k^{(l,i,j),L/U}<0} \mathbf{\Theta}_k^{(l,j),k,U/L}\Big) + \sum_{k=1}^{d_{qk}}\gamma_k^{(l,i,j),L/U}.$$

In this way, linear bounds of $\mathbf{q}^{(l,i)}(\mathbf{X})$ and $\mathbf{k}^{(l,i)}(\mathbf{X})$ are forward propagated to $\mathbf{S}_{i,j}^{(l)}$. Attention scores are normalized into attention probabilities with a softmax, i.e. $\tilde{\mathbf{S}}_{i,j}^{(l)} = \exp(\mathbf{S}_{i,j}^{(l)})/(\sum_{k=1}^n \exp(\mathbf{S}_{i,k}^{(l)}))$, where $\tilde{\mathbf{S}}_{i,j}^{(l)}$ is a normalized attention probability. $\exp(\mathbf{S}_{i,j}^{(l)})$ is an unary nonlinear function and can

be bounded by $\alpha_{i,j}^{(l),L/U}\mathbf{S}_{i,j}^{(l)}+\beta_{i,j}^{(l),L/U}$. So we forward propagate bounds of $\mathbf{S}_{i,j}^{(l)}$ to bound $\exp(\mathbf{S}_{i,j}^{(l)})$ with $\boldsymbol{\Omega}_{j,:}^{(l,i),e,L/U}\mathbf{x}^{(r)}+\boldsymbol{\Theta}_{j}^{(l,i),e,L/U}$, where:

$$\begin{cases} \boldsymbol{\Omega}_{j,:}^{(l,i),e,L/U}=\alpha_{i,j}^{(l),L/U}\boldsymbol{\Omega}_{j,:}^{(l,i),s,L/U} & \boldsymbol{\Theta}_{j}^{(l,i),e,L/U}=\alpha_{i,j}^{(l),L/U}\boldsymbol{\Theta}_{j}^{(l,i),s,L/U}+\beta_{i,j}^{(l),L/U} & \alpha_{i,j}^{(l),L/U}\geq 0, \\ \boldsymbol{\Omega}_{j,:}^{(l,i),e,L/U}=\alpha_{i,j}^{(l),L/U}\boldsymbol{\Omega}_{j,:}^{(l,i),s,U/L} & \boldsymbol{\Theta}_{j}^{(l,i),e,L/U}=\alpha_{i,j}^{(l),L/U}\boldsymbol{\Theta}_{j}^{(l,i),s,U/L}+\beta_{i,j}^{(l),L/U} & \alpha_{i,j}^{(l),L/U}< 0. \end{cases}$$

By summing up bounds of each $\exp(\mathbf{S}_{i,k}^{(l)})$, linear bounds can be further propagated to $\sum_{k=1}^{n}\exp(\mathbf{S}_{i,k}^{(l)})$. With bounds of $\exp(\mathbf{S}_{i,j}^{(l)})$ and $\sum_{k=1}^{n}\exp(\mathbf{S}_{i,k}^{(l)})$ ready, we forward propagate the bounds to $\tilde{\mathbf{S}}_{i,j}^{(l)}$ with a division similarly to bounding $\mathbf{q}_{k}^{(l,i)}(\mathbf{X})\mathbf{k}_{k}^{(l,j)}(\mathbf{X})$. The output of the self-attention $\Phi^{(l,i)}(\mathbf{X})$ is obtained with a summation of $\mathbf{v}^{(l,j)}(\mathbf{X})$ weighted by attention probability $\tilde{\mathbf{S}}_{i,k}^{(l)}$: $\Phi_{j}^{(l,i)}(\mathbf{X})=\sum_{k=1}^{n}\tilde{\mathbf{S}}_{i,k}^{(l)}\mathbf{v}_{j}^{(l,k)}(\mathbf{X})$, which can be regarded as a dot product of $\tilde{\mathbf{S}}_{i}^{(l)}$ and $\tilde{\mathbf{v}}_{k}^{(l,j)}(\mathbf{X})$, where $\tilde{\mathbf{v}}_{k}^{(l,j)}(\mathbf{X})=\mathbf{v}_{j}^{(l,k)}(\mathbf{X})$ whose bounds can be obtained from those of $\mathbf{v}_{j}^{(l,k)}(\mathbf{X})$ with a transposing. Therefore, bounds of $\tilde{\mathbf{S}}_{i,k}^{(l)}$ and $\tilde{\mathbf{v}}_{k}^{(l,j)}(\mathbf{X})$ can be forward propagated to $\Phi^{(l,i)}(\mathbf{X})$ similarly to bounding $\mathbf{S}_{i,j}^{(l)}$. In this way, we obtain the output bounds of the self-attention:

$$\boldsymbol{\Omega}_{j,:}^{(l',i),\Phi,L}\mathbf{x}^{(r)}+\boldsymbol{\Theta}_{j}^{(l',i),\Phi,L}\leq \Phi^{(l',i)}(\mathbf{X})\leq \boldsymbol{\Omega}_{j,:}^{(l',i),\Phi,U}\mathbf{x}^{(r)}+\boldsymbol{\Theta}_{j}^{(l',i),\Phi,U}. \qquad (5)$$

Recall that $\mathbf{x}^{(r)}$ is a concatenation of $\mathbf{x}^{(r_1)},\mathbf{x}^{(r_2)},\cdots,\mathbf{x}^{(r_t)}$. We can split $\boldsymbol{\Omega}_{j,:}^{(l',i),\Phi,L/U}$ into $t$ vectors with equal dimensions, $\boldsymbol{\Omega}_{j,:}^{(l',i,1),\Phi,L/U},\boldsymbol{\Omega}_{j,:}^{(l',i,2),\Phi,L/U},\cdots,\boldsymbol{\Omega}_{j,:}^{(l',i,t),\Phi,L/U}$, such that Eq. (5) becomes

$$\sum_{k=1}^{t}\boldsymbol{\Omega}_{j,:}^{(l',i,k),\Phi,L}\mathbf{x}^{(r_k)}+\boldsymbol{\Theta}_{j}^{(l',i),\Phi,L}\leq \Phi^{(l',i)}(\mathbf{X})\leq \sum_{k=1}^{t}\boldsymbol{\Omega}_{j,:}^{(l',i,k),\Phi,U}\mathbf{x}^{(r_k)}+\boldsymbol{\Theta}_{j}^{(l',i),\Phi,U}. \qquad (6)$$

**Backward Process to Self-Attention Layers** When computing bounds for a later sub-layer, the $l$-th sub-layer, using the backward process, we directly propagate the bounds at the the closest previous self-attention layer assumed to be the $l'$-th layer, to the input layer, and we skip other previous sub-layers. The bounds propagated to the $l'$-th layer are as Eq. (2). We substitute $\Phi^{(l',k)}(\mathbf{X})$ with linear bounds in Eq. (6):

$$\boldsymbol{\Lambda}_{j,:}^{(l,i,0,r_k),L/U}=\sum_{k'=1}^{n}(\boldsymbol{\Lambda}_{j,:,+}^{(l,i,l',k'),L/U}\boldsymbol{\Omega}_{j,:}^{(l',k',k),\Phi,L/U}+\boldsymbol{\Lambda}_{j,:,-}^{(l,i,l',k'),L/U}\boldsymbol{\Omega}_{j,:}^{(l',k',k),\Phi,U/L})(1\leq k\leq t),$$

$$\boldsymbol{\Lambda}_{j,:}^{(l,i,0,k),L/U}=0 \ (\forall k\notin P),$$

$$\boldsymbol{\Delta}_{j}^{(l,i,0),L/U}=\boldsymbol{\Delta}_{j}^{(l,i,l',L/U)}+\sum_{k=1}^{n}\boldsymbol{\Lambda}_{j,:,+}^{(l,i,l',k),L/U}\boldsymbol{\Theta}_{j}^{(l',k),\Phi,L/U}+\boldsymbol{\Lambda}_{j,:,-}^{(l,i,l',k),L/U}\boldsymbol{\Theta}_{j}^{(l',k),\Phi,U/L}.$$

We take global bounds as Eq. (3) and Eq. (4) to obtain the bounds of the $l$-th layer.

**Advantageous of Combining the Backward Process with a Forward Process** Introducing a forward process can significantly reduce the complexity of verifying Transformers. With the backward process only, we need to compute $\boldsymbol{\Lambda}^{(l,i,l',k)}$ and $\boldsymbol{\Delta}^{(l,i,l')}$ ($l'\leq l$), where the major cost is on $\boldsymbol{\Lambda}^{(l,i,l',k)}$ and there are $O(m^2n^2)$ such matrices to compute. The $O(n^2)$ factor is from the dependency between all pairs of positions in the input and output respectively, which makes the algorithm inefficient especially when the input sequence is long. In contrast, the forward process represents the bounds as linear functions of the perturbed positions only instead of all positions by computing $\boldsymbol{\Omega}^{(l,i)}$ and $\boldsymbol{\Theta}^{(l,i)}$. Imperceptible adversarial examples may not have many perturbed positions (Gao et al., 2018; Ko et al., 2019), and thus we may assume that the number of perturbed positions, $t$, is small. The major cost is on $\boldsymbol{\Omega}^{(l,i)}$ while there are only $O(mn)$ such matrices and the sizes of $\boldsymbol{\Lambda}^{(l,i,l',k)}$ and $\boldsymbol{\Omega}^{(l,i)}$ are relatively comparable for a small $t$. We combine the backward process and the forward process. The number of matrices $\boldsymbol{\Omega}$ in the forward process is $O(mn)$, and for the backward process, since we do not propagate bounds over self-attention layers and there is no cross-position dependency in other sub-layers, we only compute $\boldsymbol{\Lambda}^{(l,i,l',k)}$ such that $i=k$, and thus the number of matrices $\boldsymbol{\Lambda}$ is reduced to $O(m^2n)$. Therefore, the total number of matrices $\boldsymbol{\Lambda}$ and $\boldsymbol{\Omega}$ we compute is $O(m^2n)$ and is $O(n)$ times smaller than $O(m^2n^2)$ when only the backward process is used. Moreover, the backward process makes bounds tighter compared to solely the forward one, as we explain in Appendix D.

## 4 EXPERIMENTS

To demonstrate the effectiveness of our algorithm, we compute certified bounds for several sentiment classification models and perform an ablation study to show the advantage of combining the backward and forward processes. We also demonstrate the meaningfulness of our certified bounds with an application on identifying important words.

### 4.1 DATASETS AND MODELS

We use two datasets: Yelp (Zhang et al., 2015) and SST (Socher et al., 2013). Yelp consists of 560,000/38,000 examples in the training/test set and SST consists of 67,349/872/1,821 examples in the training/development/test set. Each example is a sentence or a sentence segment (for the training data of SST only) labeled with a binary sentiment polarity.

We verify the robustness of Transformers trained from scratch. For the main experiments, we consider $N$-layer models ($N \leq 3$), with 4 attention heads, hidden sizes of 256 and 512 for self-attention and feed-forward layers respectively, and we use ReLU activations for feed-forward layers. We remove the variance related terms in layer normalization, making Transformers verification bounds tighter while the clean accuracies remain comparable (see Appendix E for discussions). Although our method can be in principal applied to Transformers with any number of layers, we do not use large-scale pre-trained models such as BERT because they are too challenging to be tightly verified with the current technologies.

### 4.2 CERTIFIED BOUNDS

| Dataset | N | Acc. | $\ell_p$ | Upper | | Lower (IBP) | | Lower (Ours) | | Ours vs Upper | |
|---|---|---|---|---|---|---|---|---|---|---|---|
| | | | | Min | Avg | Min | Avg | Min | Avg | Min | Avg |
| Yelp | 1 | 91.5 | $\ell_1$ | 9.085 | 13.917 | 1.4E-4 | 3.1E-4 | 1.423 | 1.809 | 16% | 13% |
| | | | $\ell_2$ | 0.695 | 1.005 | 1.4E-4 | 3.1E-4 | 0.384 | 0.483 | 55% | 48% |
| | | | $\ell_\infty$ | 0.117 | 0.155 | 1.4E-4 | 3.1E-4 | 0.034 | 0.043 | 29% | 27% |
| | 2 | 91.5 | $\ell_1$ | 10.228 | 15.452 | 1.4E-7 | 2.2E-7 | 0.389 | 0.512 | 4% | 3% |
| | | | $\ell_2$ | 0.773 | 1.103 | 1.4E-7 | 2.2E-7 | 0.116 | 0.149 | 15% | 14% |
| | | | $\ell_\infty$ | 0.122 | 0.161 | 1.4E-7 | 2.2E-7 | 0.010 | 0.013 | 9% | 8% |
| | 3 | 91.6 | $\ell_1$ | 11.137 | 15.041 | 4.3E-10 | 7.1E-10 | 0.152 | 0.284 | 1% | 2% |
| | | | $\ell_2$ | 0.826 | 1.090 | 4.3E-10 | 7.1E-10 | 0.042 | 0.072 | 5% | 7% |
| | | | $\ell_\infty$ | 0.136 | 0.187 | 4.3E-10 | 7.1E-10 | 0.004 | 0.006 | 3% | 3% |
| SST | 1 | 83.2 | $\ell_1$ | 7.418 | 8.849 | 2.4E-4 | 2.7E-4 | 2.503 | 2.689 | 34% | 30% |
| | | | $\ell_2$ | 0.560 | 0.658 | 2.4E-4 | 2.7E-4 | 0.418 | 0.454 | 75% | 69% |
| | | | $\ell_\infty$ | 0.091 | 0.111 | 2.4E-4 | 2.7E-4 | 0.033 | 0.036 | 36% | 32% |
| | 2 | 83.5 | $\ell_1$ | 6.781 | 8.367 | 3.6E-7 | 3.8E-7 | 1.919 | 1.969 | 28% | 24% |
| | | | $\ell_2$ | 0.520 | 0.628 | 3.6E-7 | 3.8E-7 | 0.305 | 0.315 | 59% | 50% |
| | | | $\ell_\infty$ | 0.085 | 0.105 | 3.6E-7 | 3.8E-7 | 0.024 | 0.024 | 28% | 23% |
| | 3 | 83.9 | $\ell_1$ | 6.475 | 7.877 | 5.7E-10 | 6.7E-10 | 1.007 | 1.031 | 16% | 13% |
| | | | $\ell_2$ | 0.497 | 0.590 | 5.7E-10 | 6.7E-10 | 0.169 | 0.173 | 34% | 29% |
| | | | $\ell_\infty$ | 0.084 | 0.101 | 5.7E-10 | 6.7E-10 | 0.013 | 0.014 | 16% | 13% |

Table 1: Clean accuracies and computed bounds for 1-position perturbation. Bounds include upper bounds (obtained by an enumeration based method), certified lower bounds by IBP and our method respectively. We also report the gap between upper bounds and our lower bounds (represented as the percentage of lower bounds relative to upper bounds). We compute bounds for each possible option of perturbed positions and report the minimum ("Min") and average ("Avg") among them.

| | Yelp | | | | SST | | | |
|---|---|---|---|---|---|---|---|---|
| N | Lower (IBP) | | Lower (Ours) | | Lower (IBP) | | Lower (Ours) | |
| | Min | Avg | Min | Avg | Min | Avg | Min | Avg |
| 1 | 6.5E-5 | 1.2E-4 | 0.242 | 0.290 | 1.1E-4 | 1.1E-4 | 0.212 | 0.229 |
| 2 | 6.2E-8 | 8.6E-8 | 0.060 | 0.078 | 1.5E-7 | 1.5E-7 | 0.145 | 0.149 |
| 3 | 2.8E-10 | 4.4E-10 | 0.023 | 0.035 | 3.3E-10 | 4.5E-10 | 0.081 | 0.083 |

Table 2: Bounds by IBP and our method for 2-position perturbation constrained by $\ell_2$-norm.

We compute certified lower bounds for different models on different datasets. We include 1-position perturbation constrained by $\ell_1/\ell_2/\ell_\infty$-norms and 2-position perturbation constrained by $\ell_2$-norm. We compare our lower bounds with those computed by the Interval Bound Propagation (IBP) (Gowal et al., 2018) baseline. For 1-position perturbation, we also compare with *upper bounds* computed by enumerating all the words in the vocabulary and finding the word closest to the original one such that the word substitution alters the predicted label. This method has an exponential complexity with respect to the vocabulary size and can hardly be extended to perturbations on 2 or more positions; thus we do not include upper bounds for 2-position perturbation. For each example, we enumerate possible options of perturbed positions (there are $\binom{n}{t}$ options), and we integrate results from different options by taking the minimum or average respectively. We report the average results on 10 correctly classified random test examples with sentence lengths no more than 32 for 1-position perturbation and 16 for 2-position perturbation. Table 1 and Table 2 present the results for 1-position and 2-position perturbation respectively. Our certified lower bounds are significantly larger and thus tighter than those by IBP. For 1-position perturbation, the lower bounds are consistently smaller than the upper bounds, and the gap between the upper bounds and our lower bounds is reasonable compared with that in previous work on verification of feed-forward networks, e.g., in (Weng et al., 2018; Zhang et al., 2018) the upper bounds are in the order of 10 times larger than lower bounds. This demonstrates that our proposed method can compute robustness bounds for Transformers in a similar quality to the bounds of simpler neural networks.

### 4.3 Effectiveness of Combining the Backward Process with a Forward Process

| Dataset | Acc. | $\ell_p$ | Fully-Forward | | | Fully-Backward | | | Backward & Forward | | |
|---------|------|----------|------|------|------|------|------|------|------|------|------|
| | | | Min | Avg | Time | Min | Avg | Time | Min | Avg | Time |
| Yelp | 91.3 | $\ell_1$ | 2.122 | 2.173 | 12.6 | 3.485 | 3.737 | 141.4 | 3.479 | 3.729 | 24.0 |
| | | $\ell_2$ | 0.576 | 0.599 | 12.4 | 0.867 | 0.947 | 140.4 | 0.866 | 0.946 | 26.0 |
| | | $\ell_\infty$ | 0.081 | 0.084 | 12.6 | 0.123 | 0.136 | 143.9 | 0.123 | 0.136 | 26.4 |
| SST | 83.3 | $\ell_1$ | 1.545 | 1.592 | 13.7 | 1.891 | 1.961 | 177.6 | 1.891 | 1.961 | 26.5 |
| | | $\ell_2$ | 0.352 | 0.366 | 12.6 | 0.419 | 0.439 | 178.8 | 0.419 | 0.439 | 24.3 |
| | | $\ell_\infty$ | 0.048 | 0.050 | 14.6 | 0.058 | 0.061 | 181.3 | 0.058 | 0.061 | 24.3 |

Table 3: Comparison of certified lower bounds and computation time (sec) by different methods.

In the following, we show the effectiveness of combining the backward process with a forward process. We compare our proposed method (*Backward & Forward*) with two variations: 1) *Fully-Forward* propagates bounds in a forward manner for all sub-layers besides self-attention layers; 2) *Fully-Backward* computes bounds for all sub-layers including self-attention layers using the backward bound propagation and without the forward process. We compare the tightness of bounds and computation time of the three methods. We use smaller models with the hidden sizes reduced by 75%, and we use 1-position perturbation only, to accommodate *Fully-Backward* with large computational cost. Experiments are conducted on an NVIDIA TITAN X GPU. Table 3 presents the results. Bounds by *Fully-Forward* are significantly looser while those by *Fully-Backward* and *Backward & Forward* are comparable. Meanwhile, the computation time of *Backward & Forward* is significantly shorter than that of *Fully-Backward*. This demonstrates that our method of combining the backward and forward processes can compute comparably tight bounds much more efficiently.

### 4.4 Identifying Words Important to Prediction

The certified lower bounds can reflect how sensitive a model is to the perturbation of each input word. Intuitively, if a word is more important to the prediction, the model is more sensitive to its perturbation. Therefore, the certified lower bounds can be used to identify important words. In the following, we conduct an experiment to verify whether important words can be identified by our certified lower bounds. We use a 1-layer Transformer classifier under 1-position perturbation constrained by $\ell_2$-norm. The certified lower bounds are normalized by the norm of the unperturbed word embeddings respectively, when they are used for identifying the most/least important words. We compare our method with two baselines that also estimate local vulnerability: 1) *Upper* uses upper bounds; 2) *Gradient* identifies the word whose embedding has the largest $\ell_2$-norm of gradients as the most important and vice versa.

| Method | Importance Score (on SST) | Words Identified from 10 Examples on the Yelp Dataset (split by "/") |
|---|---|---|
| | | *Most Important* Words or Symbols |
| Grad | 0.47 | **terrible** / **great** / diner / **best** / **best** / food / service / food / **perfect** / **best** |
| Upper | 0.45 | **terrible** / we / . / **best** / **best** / and / **slow** / **great** / this / **best** |
| Ours | **0.57** | **terrible** / **great** / diner / **best** / **best** / **good** / **slow** / **great** / **perfect** / **best** |
| | | *Least Important* Words or Symbols |
| Grad | 0.40 | . / **decadent** / . / . / had / and / place / . / ! / . |
| Upper | 0.24 | . / . / **typical** / boba / i / dark / star / atmosphere / & / boba |
| Ours | **0.01** | . / . / . / the / . / . / food / . / . / the |

Table 4: Average importance scores of the most/least important words identified from 100 examples respectively on SST by different methods. For the *most important* words identified, larger important scores are better, and vice versa. Additionally, we show most/least important words identified from 10 examples on the Yelp dataset. Boldfaced words are considered to have strong sentiment polarities, and they should appear as most important words rather than least important ones.

**Quantitative Analysis on SST**   SST contains sentiment labels for all phrases on parse trees, where the labels range from very negative (0) to very positive (4), and 2 for neutral. For each word, assuming its label is $x$, we take $|x - 2|$, i.e., the distance to the neutral label, as the *importance score*, since less neutral words tend to be more important for the sentiment polarity of the sentence. We evaluate on 100 random test input sentences and compute the average importance scores of the most or least important words identified from the examples. In Table 4, compared to the baselines ("Upper" and "Grad"), the average importance score of the most important words identified by our lower bounds are the largest, while the least important words identified by our method have the smallest average score. This demonstrates that our method identifies the most and least important words more accurately compared to baseline methods.

**Qualitative Analysis on Yelp**   We further analyze the results on a larger dataset, Yelp. Since Yelp does not provide per-word sentiment labels, importance scores cannot be computed as on SST. Thus, we demonstrate a qualitative analysis. We use 10 random test examples and collect the words identified as the most and least important word in each example. In Table 4, most words identified as the most important by certified lower bounds are exactly the words reflecting sentiment polarities (boldfaced words), while those identified as the least important words are mostly stopwords. Baseline methods mistakenly identify more words containing no sentiment polarity as the most important. This again demonstrates that our certified lower bounds identify word importance better than baselines and our bounds provide meaningful interpretations in practice. While gradients evaluate the sensitivity of each input word, this evaluation only holds true within a very small neighborhood (where the classifier can be approximated by a first-order Taylor expansion) around the input sentence. Our certified method gives valid lower bounds that hold true within a large neighborhood specified by a perturbation set $S$, and thus it provides more accurate results.

## 5 CONCLUSION

We propose the first robustness verification method for Transformers, and tackle key challenges in verifying Transformers, including cross-nonlinearity and cross-position dependency. Our method computes certified lower bounds that are significantly tighter than those by IBP. Quantitative and qualitative analyses further show that our bounds are meaningful and can reflect the importance of different words in sentiment analysis.

## ACKNOWLEDGEMENT

This work is jointly supported by Tsinghua Scholarship for Undergraduate Overseas Studies, NSF IIS1719097 and IIS1927554, and NSFC key project with No. 61936010 and regular project with No. 61876096.

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

## A    ILLUSTRATION OF DIFFERENT BOUNDING PROCESSES

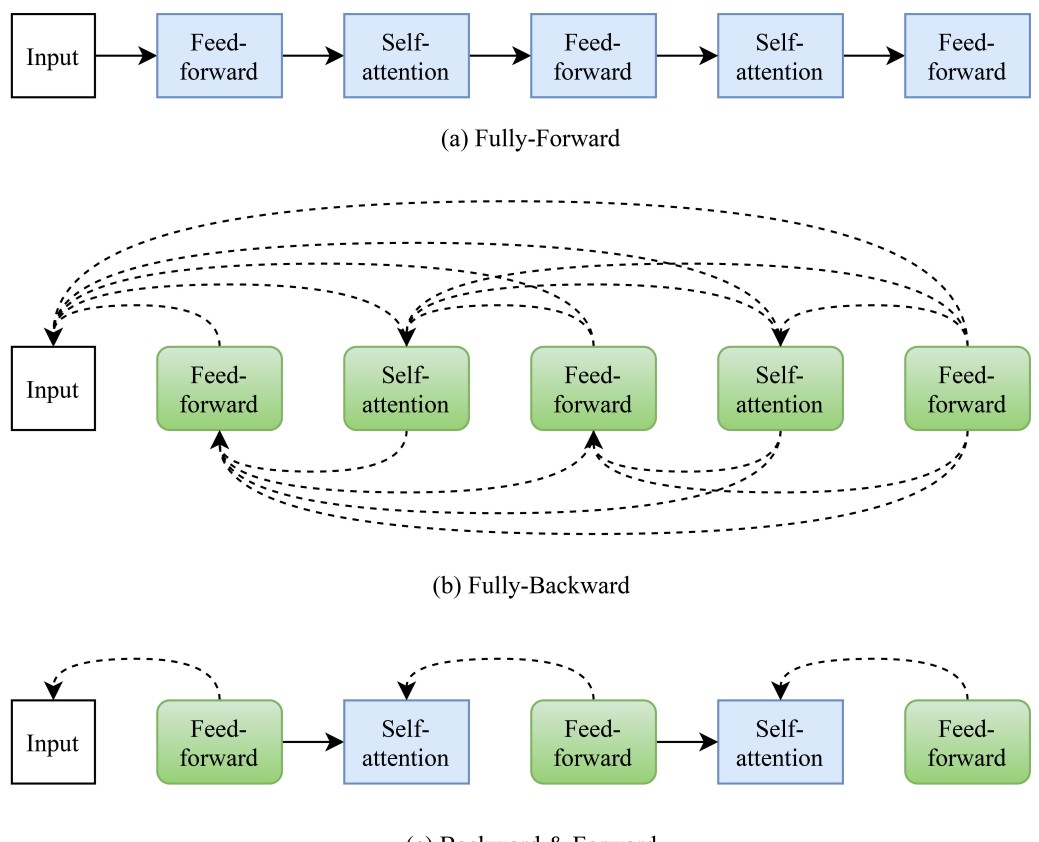

(a) Fully-Forward

(b) Fully-Backward

(c) Backward & Forward

Figure 1: Illustration of three different bounding processes: Fully-Forward (a), Fully-Backward (b), and Backward&Forward (c). We show an example of a 2-layer Transformer, where operations can be divided into two kinds of blocks, "Feed-forward" and "Self-attention". "Self-attention" contains operations in the self-attention mechanism starting from queries, keys, and values, and "Feed-forward" contains all the other operations including linear transformations and unary nonlinear functions. Arrows with solid lines indicate the propagation of linear bounds in a forward manner. Each backward arrow $A_k \rightarrow B_k$ with a dashed line for blocks $A_k, B_k$ indicates that there is a backward bound propagation to block $B_k$ when computing bounds for block $A_k$. Blocks with blue rectangles have forward processes inside the blocks, while those with green rounded rectangles have backward processes inside.

Figure 1 illustrates a comparison of the Fully-Forward, Fully-Backward and Backward & Forward processes, for a 2-layer Transformer as an example. For Fully-Forward, there are only forward processes connecting adjacent layers and blocks. For Fully-Backward, there are only backward processes, and each layer needs a backward bound propagation to all the previous layers. For our Backward & Forward algorithm, we use backward processes for the feed-forward parts and forward processes for self-attention layers, and for layers after self-attention layers, they no longer need backward bound propagation to layers prior to self-attention layers. In this way, we resolve the cross-position dependency in verifying Transformers while still keeping bounds comparably tight as those by using fully backward processes. Empirical comparison of the three frameworks are presented in Sec. 4.3.

# B  LINEAR BOUNDS OF UNARY NONLINEAR FUNCTIONS

We show in Sec. 3.2 that linear bounds can be propagated over unary nonlinear functions as long as the unary nonlinear functions can be bounded with linear functions. Such bounds are determined for each neuron respectively, according to the bounds of the input for the function. Specifically, for a unary nonlinear function $\sigma(x)$, with the bounds of $x$ obtained previously as $x \in [l, u]$, we aim to derive a linear lower bound $\alpha^L x + \beta^L$ and a linear upper bound $\alpha^U x + \beta^U$, such that

$$\alpha^L x + \beta^L \leq \sigma(x) \leq \alpha^U x + \beta^U \quad (\forall x \in [l, u]),$$

where parameters $\alpha^L, \beta^L, \alpha^U, \beta^U$ are dependent on $l, u$ and designed for different functions $\sigma(x)$ respectively. We introduce how the parameters are determined for different unary nonlinear functions involved in Transformers such that the linear bounds are valid and as tight as possible. Bounds of ReLU and tanh has been discussed by Zhang et al. (2018), and we further derive bounds of $e^x$, $\frac{1}{x}$, $x^2$, $\sqrt{x}$. $x^2$ and $\sqrt{x}$ are only used when the layer normalization is not modified for experiments to study the impact of our modification. For the following description, we define the endpoints of the function to be bounded within range $(l, r)$ as $(l, \sigma(l))$ and $(u, \sigma(u))$. We describe how the lines corresponding to the linear bounds of different functions can be determined, and thereby parameters $\alpha^L, \beta^L, \alpha^U, \beta^U$ can be determined accordingly.

**ReLU**  For ReLU activation, $\sigma(x) = \max(x, 0)$. ReLU is inherently linear on segments $(-\infty, 0]$ and $[0, \infty)$ respectively, so we make the linear bounds exactly $\sigma(x)$ for $u \leq 0$ or $l \geq 0$; and for $l < 0 < u$, we take the line passing the two endpoints as the upper bound; and we take $\sigma^L(x) = 0$ when $u < |l|$ and $\sigma^L(x) = x$ when $u \geq |l|$ as the lower bound, to minimize the gap between the lower bound and the original function.

**Tanh**  For $\tanh$ activation, $\sigma(x) = \frac{1 - e^{-2x}}{1 + e^{-2x}}$. $\tanh$ is concave for $l \geq 0$, and thus we take the line passing the two endpoints as the lower bound and take a tangent line passing $((l+u)/2, \sigma((l+u)/2)$ as the upper bound. For $u \leq 0$, $\tanh$ is convex, and thus we take the line passing the two endpoints as the upper bound and take a tangent line passing $((l+u)/2, \sigma((l+u)/2)$ as the lower bound. For $l < 0 < u$, we take a tangent line passing the right endpoint and $(d^L, \sigma(d^L))(d^L \leq 0)$ as the lower bound, and take a tangent line passing the left endpoint and $(d^U, \sigma(d^U))(d^U \geq 0)$ as the upper bound. $d^L$ and $d^U$ can be found with a binary search.

**Exp**  $\sigma(x) = exp(x) = e^x$ is convex, and thus we take the line passing the two endpoints as the upper bound and take a tangent line passing $(d, \sigma(d))$ as the lower bound. Preferably, we take $d = (l + u)/2$. However, $e^x$ is always positive and used in the softmax for computing normalized attention probabilities in self-attention layers, i.e., $exp(\mathbf{S}_{i,j}^{(l)})$ and $\sum_{k=1}^{n} exp(\mathbf{S}_{i,k}^{(l)})$. $\sum_{k=1}^{n} exp(\mathbf{S}_{i,k}^{(l)})$ appears in the denominator of the softmax, and to make reciprocal function $\frac{1}{x}$ finitely bounded, the range of $x$ should not pass 0. Therefore, we impose a constraint to force the lower bound function to be always positive, i.e., $\sigma^L(l) > 0$, since $\sigma^L(l)$ is monotonously increasing. $\sigma_d^L(x) = e^d(x - d) + e^d$ is the tangent line passing $(d, \sigma(d))$. So the constraint $\sigma_d^L(l) > 0$ yields $d < l + 1$. Hence we take $d = \min((l + u)/2, l + 1 - \Delta_d)$ where $\Delta_d$ is a small real value to ensure that $d < l + 1$ such as $\Delta_d = 10^{-2}$.

**Reciprocal**  For the reciprocal function, $\sigma(x) = \frac{1}{x}$. It is used in the softmax and layer normalization and its input is limited to have $l > 0$ by the lower bounds of $exp(x)$, and $\sqrt{x}$. With $l > 0$, $\sigma(x)$ is convex. Therefore, we take the line passing the two endpoints as the upper bound. And we take the tangent line passing $((l + u)/2, \sigma((l + u)/2))$ as the lower bound.

**Square**  For the square function, $\sigma(x) = x^2$. It is convex and we take the line passing the two endpoints as the upper bound. And we take a tangent line passing $(d, \sigma(d))(d \in [l, u])$ as the lower bound. We still prefer to take $d = (l + u)/2$. $x^2$ appears in the variance term of layer normalization and is later passed to a square root function to compute a standard derivation. To make the input to the square root function valid, i.e., non-negative, we impose a constraint $\sigma^L(x) \geq 0 (\forall x \in [l, u])$. $\sigma_d^L(x) = 2d(x - d) + d^2$ is the tangent line passing $(d, \sigma(d))$. For $u \leq 0$, $x^2$ is monotonously decreasing, the constraint we impose is equivalent to $\sigma^L(u) = 2du - d^2 \geq 0$, and with $d \leq 0$, we have $d \geq 2u$. So we take $d = \max((l + u)/2, 2u)$. For $l \geq 0$, $x^2$ is monotonously increasing, and

thus the constraint we impose is equivalent to $\sigma^L(l) = 2dl - d^2 \geq 0$, and with $d \geq 0$, we have $d \leq 2l$. So we take $d = \max((l+u)/2, 2l)$. And for $l < 0 < u$, since $\sigma^L_d(0) = -d^2$ is negative for $d \neq 0$ while $d = 0$ yields a valid lower bound, we take $d = 0$.

**Square root**  For the square root function, $\sigma(x) = \sqrt{x}$. It is used the to compute a standard derivation in layer normalization and its input is limited to be non-negative by the lower bounds of $x^2$, and thus $l \geq 0$. $\sigma(x)$ is concave, and thus we take the line passing the two endpoints as the lower bound and take the tangent line passing $((l+u)/2, \sigma((l+u)/2))$ as the upper bound.

## C   LINEAR BOUNDS OF MULTIPLICATIONS AND DIVISIONS

We provide a mathematical proof of optimal parameters for linear bounds of multiplications used in Sec. 3.3. We also show that linear bounds of division can be indirectly obtained from bounds of multiplications and the reciprocal function.

For each multiplication, we aim to bound $z = xy$ with two linear bounding planes $z^L = \alpha^L x + \beta^L y + \gamma^L$ and $z^U = \alpha^U x + \beta^U y + \gamma^U$, where $x$ and $y$ are both variables and $x \in [l_x, u_x], y \in [l_y, u_y]$ are concrete bounds of $x, y$ obtained from previous layers, such that:

$$z^L = \alpha^L x + \beta^L y + \gamma^L \leq z = xy \leq z^U = \alpha^U x + \beta^U y + \gamma^U \quad \forall (x,y) \in [l_x, u_x] \times [l_y, u_y].$$

Our goal is to determine optimal parameters of bounding planes, i.e., $\alpha^L, \beta^L, \gamma^L, \alpha^U, \beta^U, \gamma^U$, such that the bounds are as tight as possible.

### C.1   LOWER BOUND OF MULTIPLICATIONS

We define a difference function $F^L(x, y)$ which is the difference between the original function $z = xy$ and the lower bound $z^L = \alpha^L x + \beta^L y + \gamma^L$:

$$F^L(x, y) = xy - (\alpha^L x + \beta^L y + \gamma^L).$$

To make the bound as tight as possible, we aim to minimize the integral of the difference function $F^L(x, y)$ on our *concerned area* $(x, y) \in [l_x, u_x] \times [l_y, u_y]$, which is equivalent to maximizing

$$V^L = \int_{x \in [l_x, u_x]} \int_{y \in [l_y, u_y]} \alpha^L x + \beta^L y + \gamma^L, \tag{7}$$

while $F^L(x, y) \geq 0$ $(\forall (x, y) \in [l_x, u_x] \times [l_y, u_y])$. For an optimal bounding plane, there must exist a point $(x_0, y_0) \in [l_x, u_x] \times [l_y, u_y]$ such that $F^L(x_0, y_0) = 0$ (otherwise we can validly increase $\gamma^L$ to make $V^L$ larger). To ensure that $F^L(x, y) \geq 0$ within the concerned area, we need to ensure that the minimum value of $F^L(x, y)$ is non-negative. We show that we only need to check cases when $(x, y)$ is any of $(l_x, l_y), (l_x, u_y), (u_x, l_y), (u_x, u_y)$, i.e., points at the corner of the considered area. The partial derivatives of $F^L$ are:

$$\frac{\partial F^L}{\partial x} = y - \alpha^L,$$

$$\frac{\partial F^L}{\partial y} = x - \beta^L.$$

If there is $(x_1, y_1) \in (l_x, u_x) \times (l_y, u_y)$ such that $F^L(x_1, y_1) \leq F(x, y)$ $(\forall (x, y) \in [l_x, u_x] \times [l_y, u_y])$, $\frac{\partial F^L}{\partial x}(x_1, y_1) = \frac{\partial F^L}{\partial y}(x_1, y_1) = 0$ should hold true. Thereby $\frac{\partial F^L}{\partial x}(x, y), \frac{\partial F^L}{\partial y}(x, y) < 0$ $(\forall (x, y) \in [l_x, x_1) \times [l_y, y_1))$, and thus $F^L(l_x, l_y) < F^L(x_1, y_1)$ and $(x_1, y_1)$ cannot be the point with the minimum value of $F^L(x, y)$. On the other hand, if there is $(x_1, y_1)(x_1 = l_x, y_1 \in (l_y, u_y))$, i.e., on one border of the concerned area but not on any corner, $\frac{\partial F^L}{\partial y}(x_1, y_1) = 0$ should hold true. Thereby, $\frac{\partial F^L}{\partial y}(x, y) = \frac{\partial F^L}{\partial y}(x_1, y) = 0$ $(\forall (x, y), x = x_1 = l_x)$, and $F^L(x_1, y_1) = F^L(x_1, l_y) = F^L(l_x, l_y)$. This property holds true for the other three borders of the concerned area. Therefore, other points within the concerned area cannot have smaller function value $F^L(x, y)$, so we only

need to check the corners, and the constraints on $F^L(x, y)$ become

$$
\begin{cases}
F^L(x_0, y_0) & = 0 \\
F^L(l_x, l_y) & \geq 0 \\
F^L(l_x, u_y) & \geq 0 \\
F^L(u_x, l_y) & \geq 0 \\
F^L(u_x, u_y) & \geq 0
\end{cases},
$$

which is equivalent to

$$
\begin{cases}
\gamma^L = x_0 y_0 - \alpha^L x_0 - \beta^L y_0 \\
l_x l_y - \alpha^L(l_x - x_0) - \beta^L(l_y - y_0) - x_0 y_0 \geq 0 \\
l_x u_y - \alpha^L(l_x - x_0) - \beta^L(u_y - y_0) - x_0 y_0 \geq 0 \\
u_x l_y - \alpha^L(u_x - x_0) - \beta^L(l_y - y_0) - x_0 y_0 \geq 0 \\
u_x u_y - \alpha^L(u_x - x_0) - \beta^L(u_y - y_0) - x_0 y_0 \geq 0
\end{cases}.
\tag{8}
$$

We substitute $\gamma^L$ in Eq. (7) with Eq. (8), yielding

$$
V^L = V_0[(l_x + u_x - 2x_0)\alpha^L + (l_y + u_y - 2y_0)\beta^L + 2x_0 y_0],
$$

where $V_0 = \frac{(u_x - l_x)(u_y - l_y)}{2}$.

We have shown that the minimum function value $F^L(x, y)$ within the concerned area cannot appear in $(l_x, u_x) \times (l_y, u_y)$, i.e., it can only appear at the border. When $(x_0, y_0)$ is a point with a minimum function value $F^L(x_0, y_0) = 0$, $(x_0, y_0)$ can also only be chosen from the border of the concerned area. At least one of $x_0 = l_x$ and $x_0 = u_x$ holds true.

If we take $x_0 = l_x$:

$$
V_1^L = V_0[(u_x - l_x)\alpha^L + (l_y + u_y - 2y_0)\beta^L + 2l_x y_0].
$$

And from Eq. (8) we obtain

$$
\alpha^L \leq \frac{u_x l_y - l_x y_0 - \beta^L(l_y - y_0)}{u_x - l_x},
$$

$$
\alpha^L \leq \frac{u_x u_y - l_x y_0 - \beta^L(u_y - y_0)}{u_x - l_x},
$$

$$
l_x \leq \beta^L \leq l_x \Leftrightarrow \beta^L = l_x.
$$

Then

$$
V_1^L = V_0[(u_x - l_x)\alpha^L + l_x(l_y + u_y)],
$$

$$
\begin{aligned}
(u_x - l_x)\alpha^L &\leq -l_x y_0 + \min(u_x l_y - \beta^L(l_y - y_0), u_x u_y - \beta^L(u_y - y_0)) \\
&= -l_x y_0 + \min(u_x l_y - l_x(l_y - y_0), u_x u_y - l_x(u_y - y_0)) \\
&= (u_x - l_x)\min(l_y, u_y) \\
&= (u_x - l_x)l_y.
\end{aligned}
$$

Therefore,

$$
\alpha^L \leq l_y.
$$

To maximize $V_1^L$, since now only $\alpha^L$ is unknown in $V_1^L$ and the coefficient of $\alpha^L$ is $V_0(u_x - l_x) \geq 0$, we take $\alpha^L = l_y$, and then

$$
V_1^L = V_0(u_x l_y + l_x u_y)
$$

is a constant.

For the other case if we take $x_0 = u_x$:

$$
V_2^L = V_0[(l_x - u_x)\alpha^L + (l_y + u_y - 2y_0)\beta^L + 2u_x y_0],
$$

$$
\alpha^L \geq \frac{l_x l_y - u_x y_0 - \beta^L(l_y - y_0)}{l_x - u_x},
$$

$$\alpha^L \geq \frac{l_x u_y - u_x y_0 - \beta^L(u_y - y_0)}{l_x - u_x},$$

$$u_x \leq \beta^L \leq u_x \Leftrightarrow \beta^L = u_x,$$

$$V_2^L = V_0[(l_x - u_x)\alpha^L + u_x(l_y + u_y)],$$

$$
\begin{aligned}
(l_x - u_x)\alpha^L &\leq -u_x y_0 + \min(l_x l_y - \beta^L(l_y - y_0), l_x u_y - \beta^L(u_y - y_0)) \\
&= \min(l_x l_y - u_x l_y, l_x u_y - u_x u_y) \\
&= (l_x - u_x)\max(l_y, u_y) \\
&= (l_x - u_x)u_y.
\end{aligned}
$$

Therefore,
$$\alpha^L \geq u_y.$$

We take $\alpha^L = u_y$ similarly as in the case when $x_0 = l_x$, and then

$$V_2^L = V_0(l_x u_y + u_x l_y).$$

We notice that $V_1^L = V_2^L$, so we can simply adopt the first one. We also notice that $V_1^L, V_2^L$ are independent of $y_0$, so we may take any $y_0$ within $[l_y, u_y]$ such as $y_0 = l_y$. Thereby, we obtain the a group of optimal parameters of the lower bounding plane:

$$
\begin{cases}
\alpha^L &= l_y \\
\beta^L &= l_x \\
\gamma^L &= -l_x l_y
\end{cases}.
$$

## C.2 UPPER BOUND OF MULTIPLICATIONS

We derive the upper bound similarly. We aim to minimize

$$V^U = V_0[(l_x + u_x - 2x_0)\alpha^U + (l_y + u_y - 2y_0)\beta^U + 2x_0 y_0],$$

where $V_0 = \frac{(u_x - l_x)(u_y - l_y)}{2}$.

If we take $x_0 = l_x$:

$$V_1^U = V_0[(u_x - l_x)\alpha^U + (l_y + u_y - 2y_0)\beta^U + 2l_x y_0],$$

$$\alpha^U \geq \frac{u_x l_y - l_x y_0 - \beta^U(l_y - y_0)}{u_x - l_x},$$

$$\alpha^U \geq \frac{u_x u_y - l_x y_0 - \beta^U(u_y - y_0)}{u_x - l_x},$$

$$l_x \leq \beta^U \leq l_x \Leftrightarrow \beta^U = l_x.$$

Then
$$V_1^U = V_0[(u_x - l_x)\alpha^U + l_x(l_y + u_y)],$$

$$
\begin{aligned}
(u_x - l_x)\alpha^U &\geq -l_x y_0 + \max(u_x l_y - \beta^U(l_y - y_0), u_x u_y - \beta^U(u_y - y_0)) \\
&= \max(u_x l_y - l_x l_y, u_x u_y - l_x u_y) \\
&= (u_x - l_x)\max(l_y, u_y) \\
&= (u_x - l_x)u_y.
\end{aligned}
$$

Therefore,
$$\alpha^U \geq u_y.$$

To minimize $V_1^U$, we take $\alpha^U = u_y$, and then

$$V_1^U = V_0(l_x l_y + u_x u_y).$$

For the other case if we take $x_0 = u_x$:

$$V_2^U = V_0[(l_x - u_x)\alpha^U + (l_y + u_y - 2y_0)\beta^U + 2u_x y_0],$$

$$\alpha^U \leq \frac{l_x l_y - u_x y_0 - \beta^U(l_y - y_0)}{l_x - u_x},$$

$$\alpha^U \leq \frac{l_x u_y - u_x y_0 - \beta^U(u_y - y_0)}{l_x - u_x},$$

$$u_x \leq \beta^U \leq u_x \Leftrightarrow \beta^U = u_x.$$

Therefore,

$$V_2^U = V_0[(l_x - u_x)\alpha^U + u_x(l_y + u_y)],$$

$$(l_x - u_x)\alpha^U \geq -u_x y_0 + \max(l_x l_y - \beta^U(l_y - y_0), l_x u_y - \beta^U(u_y - y_0))$$
$$= \max(l_x l_y - u_x l_y, l_x u_y - u_x u_y)$$
$$= (l_x - u_x)\min(l_y, u_y)$$
$$= (l_x - u_x)l_y.$$

Therefore,

$$\alpha^U \leq l_y.$$

To minimize $V_2^U$, we take $\alpha^U = l_y$, and then

$$V_2^U = V_0(l_x l_y + u_x u_y).$$

Since $V_1^U = V_2^U$, we simply adopt the first case. And $V_1^U, V_2^U$ are independent of $y_0$, so we may take any $y_0$ within $[l_y, u_y]$ such as $y_0 = l_y$. Thereby, we obtain a group of optimal parameters of the upper bounding plane:

$$\begin{cases} \alpha^U &= u_y \\ \beta^U &= l_x \\ \gamma^U &= -l_x u_y \end{cases}.$$

### C.3 LINEAR BOUNDS OF DIVISIONS

We have shown that closed-form linear bounds of multiplications can be derived. However, we find that directly bounding $z = \frac{x}{y}$ is relatively more difficult. If we try to derive a lower bound $z^L = \alpha^L x + \beta^L y + \gamma^L$ for $z = \frac{x}{y}$ as shown in Appendix C.1, the difference function is

$$F^L(x, y) = \frac{x}{y} - (\alpha^L x + \beta^L y + \gamma^L).$$

It is possible that a minimum function value of $F^L(x, y)$ for $(x, y)$ within the concerned area appears at a point other than the corners. For example, for $l_x = 0.05, u_x = 0.15, l_y = 0.05, u_y = 0.15, \alpha = 10, \beta = -20, \gamma = 2$, the minimum function value of $F^L(x, y)$ for $(x, y) \in [0.05, 0.15] \times [0.05, 0.15]$ appears at $(0.1, 0.1)$ which is not a corner of $[0.05, 0.15] \times [0.05, 0.15]$. This makes it more difficult to derive closed-form parameters such that the constraints on $F^L(x, y)$ are satisfied. Fortunately, we can bound $z = \frac{x}{y}$ indirectly by utilizing the bounds of multiplications and reciprocal functions. We bound $z = \frac{x}{y}$ by first bounding a unary function $\bar{y} = \frac{1}{y}$ and then bounding the multiplication $z = x\bar{y}$.

## D TIGHTNESS OF BOUNDS BY THE BACKWARD PROCESS AND FORWARD PROCESS

We have discussed that combining the backward process with a forward process can reduce computational complexity, compared to the method with the backward process only. But we only use the forward process for self-attention layers and do not fully use the forward process for all sublayers, because bounds by the forward process can be looser than those by the backward process.

We compare the tightness of bounds by the forward process and the backward process respectively. To illustrate the difference, for simplicity, we consider a $m$-layer feed-forward network $\Phi^{(0)} = \mathbf{x}$, $\mathbf{y}^{(l)} = \mathbf{W}^{(l)}\Phi^{(l-1)}(\mathbf{x}) + \mathbf{b}^{(l)}$, $\Phi^{(l)}(\mathbf{x}) = \sigma(\mathbf{y}^{(l)}(\mathbf{x}))(0 < l \leq m)$, where $\mathbf{x}$ is the input vector, $\mathbf{W}^{(l)}$ and $\mathbf{b}^{(l)}$ are the weight matrix and the bias vector for the $l$-th layer respectively, $\mathbf{y}^{(l)}(\mathbf{x})$ is the pre-activation vector of the $l$-th layer, $\Phi^{(l)}(\mathbf{x})$ is the vector of neurons in the $l$-th layer, and $\sigma(\cdot)$ is an activation function. Before taking global bounds, both the backward process and the forward process bound $\Phi_j^{(l)}(\mathbf{x})$ with linear functions of $\mathbf{x}$. When taking global bounds as Eq. (3) and Eq. (4), only the norm of weight matrix is directly related to the $\epsilon$ in binary search for certified lower bounds. Therefore, we try to measure the tightness of the computed bounds using the difference between weight matrices for lower bounds and upper bounds respectively. We show how it is computed for the forward process and the backward process respectively.

## D.1 THE FORWARD PROCESS

For the forward process, we bound each neuron $\Phi_j^{(l)}(\mathbf{x})$ with linear functions:

$$\boldsymbol{\Omega}_{j,:}^{(l),L}\mathbf{x} + \boldsymbol{\Theta}_j^{(l),L} \leq \Phi_j^{(l)}(\mathbf{x}) \leq \boldsymbol{\Omega}_{j,:}^{(l),U}\mathbf{x} + \boldsymbol{\Theta}_j^{(l),U}.$$

To measure the tightness of the bounds, we are interested in $\boldsymbol{\Omega}^{(l),L}$, $\boldsymbol{\Omega}^{(l),U}$, and also $\boldsymbol{\Omega}^{(l),U} - \boldsymbol{\Omega}^{(l),L}$. Initially,

$$\boldsymbol{\Omega}^{(0),L/U} = \mathbf{I}, \ \boldsymbol{\Theta}^{(0),L/U} = \mathbf{0}, \ \boldsymbol{\Omega}^{(0),U} - \boldsymbol{\Omega}^{(0),L} = \mathbf{0}.$$

We can forward propagate the bounds of $\Phi^{(l-1)}(\mathbf{x})$ to $\mathbf{y}^{(l)}(\mathbf{x})$:

$$\boldsymbol{\Omega}_{j,:}^{(l),y,L}\mathbf{x} + \boldsymbol{\Theta}_j^{(l),y,L} \leq \mathbf{y}_j^{(l)}(\mathbf{x}) \leq \boldsymbol{\Omega}_{j,:}^{(l),y,U}\mathbf{x} + \boldsymbol{\Theta}_j^{(l),y,U},$$

where

$$\boldsymbol{\Omega}_{j,:}^{(l),y,L/U} = \sum_{\mathbf{W}_{j,i}^{(l)}>0} \mathbf{W}_{j,i}^{(l)}\boldsymbol{\Omega}_{i,:}^{(l-1),L/U} + \sum_{\mathbf{W}_{j,i}^{(l)}<0} \mathbf{W}_{j,i}^{(l)}\boldsymbol{\Omega}_{i,:}^{(l-1),U/L},$$

$$\boldsymbol{\Theta}^{(l),y,L/U} = \sum_{\mathbf{W}_{j,i}^{(l)}>0} \mathbf{W}_{j,i}^{(l)}\boldsymbol{\Theta}_i^{(l-1),L/U} + \sum_{\mathbf{W}_{j,i}^{(l)}<0} \mathbf{W}_{j,i}^{(l)}\boldsymbol{\Theta}_i^{(l-1),U/L} + \mathbf{b}^{(l)}.$$

With the global bounds of $\mathbf{y}^{(l)}(\mathbf{x})$ that can be obtained with Eq. (3) and Eq. (4), we bound the activation function:

$$\alpha_j^{(l),L}\mathbf{y}^{(l)}(\mathbf{x}) + \beta_j^{(l),L} \leq \sigma(\mathbf{y}_j^{(l)}(\mathbf{x})) \leq \alpha_j^{(l),U}\mathbf{y}^{(l)}(\mathbf{x}) + \beta_j^{(l),U}.$$

And then bounds can be propagated from $\Phi^{(l-1)}(\mathbf{x})$ to $\Phi^{(l)}(\mathbf{x})$:

$$\boldsymbol{\Omega}_{j,:}^{(l),L/U} = \begin{cases} \alpha_j^{(l),L/U}\boldsymbol{\Omega}_{j,:}^{(l),y,L/U} & \alpha_j^{(l),L/U} \geq 0 \\ \alpha_j^{(l),L/U}\boldsymbol{\Omega}_{j,:}^{(l),y,U/L} & \alpha_j^{(l),L/U} < 0 \end{cases},$$

$$\boldsymbol{\Theta}_j^{(l),L/U} = \begin{cases} \alpha_j^{(l),L/U}\boldsymbol{\Theta}_j^{(l),y,L/U} + \beta_j^{(l),L/U} & \alpha_j^{(l),L/U} \geq 0 \\ \alpha_j^{(l),L/U}\boldsymbol{\Theta}_j^{(l),y,U/L} + \beta_j^{(l),L/U} & \alpha_j^{(l),L/U} < 0 \end{cases}.$$

Therefore,

$$\boldsymbol{\Omega}_{j,:}^{(l),U} - \boldsymbol{\Omega}_{j,:}^{(l),L} = (\alpha_j^{(l),U} - \alpha_j^{(l),L})|\mathbf{W}_j^{(l)}|(\boldsymbol{\Omega}_{j,:}^{(l-1),U} - \boldsymbol{\Omega}_{j,:}^{(l-1),L}) \tag{9}$$

illustrates how the tightness of the bounds is changed from earlier layers to later layers.

## D.2 THE BACKWARD PROCESS AND DISCUSSIONS

For the backward process, we bound the neurons in the $l$-th layer with linear functions of neurons in a previous layer, the $l'$-th layer:

$$\Phi^{(l,l'),L} = \boldsymbol{\Lambda}_{j,:}^{(l,l'),L}\Phi^{(l')}(\mathbf{x}) + \boldsymbol{\Delta}_j^{(l,l'),L} \leq \Phi^{(l)}(\mathbf{x}) \leq \boldsymbol{\Lambda}_{j,:}^{(l,l'),U}\Phi^{(l')}(\mathbf{x}) + \boldsymbol{\Delta}_j^{(l,l'),U} = \Phi^{(l,l'),U}.$$

We have shown in Sec. 3.1 how such bounds can be propagated to $l' = 0$, for the case when the input is sequential. For the nonsequential case we consider here, it can be regarded as a special case when the input length is 1. So we can adopt the method in Sec. 3.1 to propagate bounds

for the feed-forward network we consider here. We are interested in $\mathbf{\Lambda}^{(l,l'),L}$, $\mathbf{\Lambda}^{(l,l'),U}$ and also $\mathbf{\Lambda}^{(l,l'),U} - \mathbf{\Lambda}^{(l,l'),L}$. Weight matrices of linear bounds before taking global bounds are $\mathbf{\Lambda}^{(l,0),L}$ and $\mathbf{\Lambda}^{(l,0),U}$ which are obtained by propagating the bounds starting from $\mathbf{\Lambda}^{(l,l),L} = \mathbf{\Lambda}^{(l,l),U} = \mathbf{I}$. According to bound propagation described in Sec. 3.2,

$$\mathbf{\Lambda}^{(l,l'-1),U}_{:,j} - \mathbf{\Lambda}^{(l,l'-1),L}_{:,j} = (\alpha_j^{(l'),U}(\mathbf{\Lambda}^{(l,l'),U}_{:,j,+} - \mathbf{\Lambda}^{(l,l'),L}_{:,j,-}) - \alpha_j^{(l'),L}(\mathbf{\Lambda}^{(l,l'),L}_{:,j,+} - \mathbf{\Lambda}^{(l,l'),U}_{:,j,-}))\mathbf{W}^{(l')} \quad (10)$$

illustrates how the tightness bounds can be measured during the backward bound propagation until $l' = 0$.

There is a $\mathbf{W}^{(l')}$ in Eq. (10) instead of $|\mathbf{W}^{(l')}|$ in Eq. (9). The norm of $(\mathbf{\Omega}^{(l),U}_{j,:} - \mathbf{\Omega}^{(l),L}_{j,:})$ in Eq. (9) can quickly grow large as $l$ increases during the forward propagation when $\|\mathbf{W}^{(l)}_j\|$ is greater than 1, while this generally holds true for neural networks to have $\|\mathbf{W}^{(l)}_j\|$ greater than 1 in feed-forward layers. While in Eq. (10), $\mathbf{W}^{(l')}_j$ can have both positive and negative elements and tends to allow cancellations for different $\mathbf{W}^{(l')}_{j,i}$, and thus the norm of $(\mathbf{\Lambda}^{(l,l'-1),U}_{:,j} - \mathbf{\Lambda}^{(l,l'-1),L}_{:,j})$ tends to be smaller. Therefore, the bounds computed by the backward process tend to be tighter than those by the forward framework, which is consistent with our experiment results in Table 3.

## E  IMPACT OF MODIFYING THE LAYER NORMALIZATION

The original Transformers have a layer normalization after the embedding layer, and two layer normalization before and after the feed-forward part respectively in each Transformer layer. We modify the layer normalization, $f(\mathbf{x}) = \mathbf{w}(\mathbf{x} - \mu)/\sigma + \mathbf{b}$, where $\mathbf{x}$ is $d$-dimensional a vector to be normalized, $\mu$ and $\sigma$ are the mean and standard derivation of $\{\mathbf{x}_i\}$ respectively, and $\mathbf{w}$ and $\mathbf{b}$ are gain and bias parameters respectively. $\sigma = \sqrt{(1/d)\sum_{i=1}^{d}(\mathbf{x}_i - \mu)^2 + \epsilon_s}$ where $\epsilon_s$ is a smoothing constant. It involves $(\mathbf{x}_i - \mu)^2$ whose linear lower bound is loose and exactly 0 when the range of the $\mathbf{x}_i - \mu$ crosses 0. When the $\ell_p$-norm of the perturbation is relatively larger, there can be many $\mathbf{x}_i - \mu$ with ranges crossing 0, which can cause the lower bound of $\sigma$ to be small and thereby the upper bound of $f_i(\mathbf{x})$ to be large. This can make the certified bounds loose. To tackle this, we modify the layer normalization into $f(\mathbf{x}) = \mathbf{w}(\mathbf{x} - \mu) + \mathbf{b}$ by removing the standard derivation term. We use an experiment to study the impact of this modification. We compare the clean accuracies and certified bounds of the models with modified layer normalization to models with standard layer normalization and with no layer normalization respectively. Table 5 presents the results. Certified lower bounds of models with no layer normalization or our modification are significantly tighter than those of corresponding models with the standard layer normalization. Meanwhile, the clean accuracies of the models with our modification are comparable with those of the models with the standard layer normalization (slightly lower on Yelp and slightly higher on SST). This demonstrates that it appears to be worthwhile to modify the layer normalization in Transformers for easier verification.

| Dataset | $N$ | LayerNorm | Acc. | $\ell_p$ | Upper Min | Upper Avg | Lower Min | Lower Avg | Ours vs Upper Min | Ours vs Upper Avg |
|---|---|---|---|---|---|---|---|---|---|---|
| Yelp | 1 | Standard | 91.7 | $\ell_1$ | 189.934 | 199.265 | 0.010 | 0.022 | 5.3E-5 | 1.1E-4 |
| | | | | $\ell_2$ | 15.125 | 15.384 | 0.008 | 0.019 | 5.5E-4 | 1.3E-3 |
| | | | | $\ell_\infty$ | 2.001 | 3.066 | 0.002 | 0.005 | 8.7E-4 | 1.7E-3 |
| | | None | 91.4 | $\ell_1$ | 8.044 | 12.948 | 1.360 | 1.684 | 17% | 13% |
| | | | | $\ell_2$ | 0.580 | 0.905 | 0.363 | 0.447 | 63% | 49% |
| | | | | $\ell_\infty$ | 0.086 | 0.127 | 0.033 | 0.040 | 38% | 32% |
| | | Ours | 91.5 | $\ell_1$ | 9.085 | 13.917 | 1.423 | 1.809 | 16% | 13% |
| | | | | $\ell_2$ | 0.695 | 1.005 | 0.384 | 0.483 | 55% | 48% |
| | | | | $\ell_\infty$ | 0.117 | 0.155 | 0.034 | 0.043 | 29% | 27% |
| | 2 | Standard | 92.0 | $\ell_1$ | 190.476 | 201.092 | 0.002 | 0.004 | 1.2E-5 | 1.8E-5 |
| | | | | $\ell_2$ | 15.277 | 15.507 | 0.001 | 0.002 | 9.0E-5 | 1.6E-4 |
| | | | | $\ell_\infty$ | 2.022 | 2.901 | 0.000 | 0.000 | 9.5E-5 | 1.3E-4 |
| | | None | 91.5 | $\ell_1$ | 8.112 | 15.225 | 0.512 | 0.631 | 6% | 4% |
| | | | | $\ell_2$ | 0.587 | 1.042 | 0.123 | 0.154 | 21% | 15% |
| | | | | $\ell_\infty$ | 0.081 | 0.140 | 0.010 | 0.013 | 13% | 9% |
| | | Ours | 91.5 | $\ell_1$ | 10.228 | 15.452 | 0.389 | 0.512 | 4% | 3% |
| | | | | $\ell_2$ | 0.773 | 1.103 | 0.116 | 0.149 | 15% | 14% |
| | | | | $\ell_\infty$ | 0.122 | 0.161 | 0.010 | 0.013 | 9% | 8% |
| SST | 1 | Standard | 83.0 | $\ell_1$ | 190.777 | 194.961 | 0.008 | 0.015 | 4.2E-5 | 7.8E-5 |
| | | | | $\ell_2$ | 15.549 | 15.630 | 0.006 | 0.013 | 4.1E-4 | 8.2E-4 |
| | | | | $\ell_\infty$ | 2.241 | 2.504 | 0.001 | 0.003 | 5.2E-4 | 1.2E-3 |
| | | None | 83.0 | $\ell_1$ | 6.921 | 8.417 | 2.480 | 2.659 | 36% | 32% |
| | | | | $\ell_2$ | 0.527 | 0.628 | 0.411 | 0.447 | 78% | 71% |
| | | | | $\ell_\infty$ | 0.089 | 0.109 | 0.032 | 0.035 | 36% | 32% |
| | | Ours | 83.2 | $\ell_1$ | 7.418 | 8.849 | 2.503 | 2.689 | 34% | 30% |
| | | | | $\ell_2$ | 0.560 | 0.658 | 0.418 | 0.454 | 75% | 69% |
| | | | | $\ell_\infty$ | 0.091 | 0.111 | 0.033 | 0.036 | 36% | 32% |
| | 2 | Standard | 82.5 | $\ell_1$ | 191.742 | 196.365 | 0.002 | 0.004 | 9.6E-6 | 1.9E-5 |
| | | | | $\ell_2$ | 15.554 | 15.649 | 0.001 | 0.003 | 7.0E-5 | 1.7E-4 |
| | | | | $\ell_\infty$ | 2.252 | 2.513 | 0.000 | 0.000 | 6.6E-5 | 1.9E-4 |
| | | None | 83.5 | $\ell_1$ | 6.742 | 8.118 | 1.821 | 1.861 | 27% | 23% |
| | | | | $\ell_2$ | 0.515 | 0.610 | 0.298 | 0.306 | 58% | 50% |
| | | | | $\ell_\infty$ | 0.085 | 0.103 | 0.023 | 0.024 | 28% | 23% |
| | | Ours | 83.5 | $\ell_1$ | 6.781 | 8.367 | 1.919 | 1.969 | 28% | 24% |
| | | | | $\ell_2$ | 0.520 | 0.628 | 0.305 | 0.315 | 59% | 50% |
| | | | | $\ell_\infty$ | 0.085 | 0.105 | 0.024 | 0.024 | 28% | 23% |

Table 5: Clean accuracies, upper bounds, certified lower bounds by our method of models with different layer normalization settings.

