# OpenReview forum: "Robustness Verification for Transformers"
_ICLR.cc/2020/Conference — Accept (Poster)_

### Official Review · AnonReviewer2 · 2019-10-22
**Official Blind Review #2**

**Rating:** 3

**Review:**

Overview:

This paper is dedicated to developing robustness verification techniques. They claim their methods can deal with verification problems for Transformers which includes cross-nonlinearity and cross-position dependency. The paper solves these key challenges which are not traceable for previous methods. Moreover, the author demonstrates their certified robustness bounds are significantly higher than those by naive Interval Bound Propagation. They also point out the practice meaning through sentiment analysis.

Strength Bullets:

1. It is an interesting design that combines the backward process with a forward process. The author also conducts an ablation experiment to show its advantages both in the bound estimation and computation time cost.
2. The derivation is very solid and detailed. The author not only gives the certified bounds but also further analyzes whether the bounds are reasonable.

Weakness Bullets:

1. For comparison experiments, especially in Table 2, the author doesn't compare with other previous state-of-the-art methods. It just an ablation among fully forward, fully backward and combine two. To be more convincing, the author needs to post not only bounds but also time costs cross several methods on one or two datasets.
2. The experiment only uses a single-layer transformer, I don't think it contains much more nonlinear operations then MLP, CNN or RNN in the previous work. The author needs to add experiments of multi-layer transformers.
3. The author doesn't list all famous previous verification framework, like MILP (EVALUATING ROBUSTNESS OF NEURAL NETWORKS WITH MIXED INTEGER PROGRAMMING), even doesn't give a cite. MILP is a powerful verification technique that can deal with cross-position dependency, which can be potentially applied to transformers. The author needs to compare these methods to prove the advantages of the paper's approach.
4. The novelty of the main contribution of the paper is arguable. Although it is the first one who does robustness verification on transformers, the linear relaxation is similar to previous work just deal with different nonlinearity. The author may provide more detail and explanation about the creative modification in relaxation or other parts of the proposed verification framework.

Recommendation:

For lack of necessary experiments and limit novelty, even if I like part of the approach design, this is a weak reject.

**Experience Assessment:**

I have read many papers in this area.

**Review Assessment: Checking Correctness Of Derivations And Theory:**

I assessed the sensibility of the derivations and theory.

**Review Assessment: Checking Correctness Of Experiments:**

I assessed the sensibility of the experiments.

**Review Assessment: Thoroughness In Paper Reading:**

I read the paper at least twice and used my best judgement in assessing the paper.

---

> ### Author Response · Authors · 2019-11-13
> **We have results for both 1-layer and 2-layer Transformers and we have further added new results on 3-layer Transformers.**
>
> We thank the reviewer for the constructive feedback. We address the concern regarding multi-layer Transformers below:
>
> Our initial submission included results for multi-layer transformers. As you can see in Table 1, a column called “N” indicates the number of layers. We had experiments for both 1-layer and 2-layer Transformers. We have also further added experiments for 3-layer Transformers in the revised paper.
>
> Each Transformer layer actually contains many nonlinear operations: dot product operations, softmax, weighted summation, and also element-wise activation functions in the feed-forward layers after self-attention. Especially, bivariate nonlinearity in dot product and weighted summation are much more complicated than nonlinearities in a MLP/CNN/RNN layer (e.g. a MLP layer is just a linear operation plus an element-wise activation function). Therefore, we believe that even a *single* Transformer layer already has an amount of nonlinear operations comparable to those in many MLP/CNN/RNN layers. We believe that the provided up to 3-layer Transformer results in Table 1 should be sufficient to demonstrate the ability of our algorithm in the multi-layer setting.

---

> ### Author Response · Authors · 2019-11-13
> **Discussions on comparisons to SOTA methods and our novelty**
>
> We thank the reviewer for the constructive feedback. We address point 1, 2, and 4 in “weakness bullets” below (we have addressed point 3 in another reply).
>
> (1) Regarding comparison with previous SOTA methods, and time cost comparison:
>
> Because we propose the first algorithm for verifying Transformers, no previous work is directly applicable for comparison. Previous methods were specifically designed for simpler neural networks such as feed-forward networks or RNNs. They cannot be extended to Transformer in a straightforward manner, except for naive IBP (Gowal et al., 2018). We have adapted IBP to verifying Transformers as a baseline in the paper.  MILP can hardly be extended to Transformers (you may refer to our response to your question regarding MILP below).
>
> We have already included time costs for fully forward, fully backward and our “backward & forward” methods (Table 2 in the initial draft and Table 3 in the revised one). We think it is unnecessary to compare time costs with IBP, because IBP cannot produce non-vacuous bounds for Transformers at all (lower bounds computed by IBP are extremely small; see Table 1).
>
> (2) Regarding citations and MILP:
>
> We thank the reviewer for raising the concern of listing previous verification works. In our revised paper, we have improved the introduction and related work sections and included many related papers that we missed, and also fixed some inaccurate citations.
>
> We cannot compare to MILP based methods (Tjeng et al., 2019; Dutta et al., 2018), as they mainly focus on ReLU non-linearity. It is non-trivial to extend MILP based method to more complicated functions like self-attention, as the MILP formulation crucially depends on the property of ReLU (or other piecewise linear) neurons: ReLU neurons have an “active” state that passes the input, and an “inactive” state that outputs a constant 0. This can be formulated as a binary variable in MILP (see Eq. (6) in Tjeng et al., 2019), but cannot be easily extended to self-attention layers with complex cross-nonlinearity and non-piecewise linear functions (dot product, weighted sum, softmax, tanh, etc).
>
> Additionally, since MILP searches the optimal solution for the non-convex verification problem (Eq. 1), they are significantly slower. Weng et al., 2018 showed that similar methods can be over 10,000 times slower than backward bounding; Salman et al., 2019 had to use a cluster with 1000 CPU nodes to conduct MILP verification for simple CNN networks. So we believe MILP cannot scale to the transformer verification problem at all. Our method efficiently finds a lower bound instead, similar to CROWN (Zhang et al., 2018) and other efficient verifiers.
>
> (3) Regarding our novelty:
>
> Our work is the first work on verifying Transformers. Previous methods were specifically designed for simple architectures such as feed-forward networks and RNNs, and they cannot be directly applied to Transformers with a self-attention architecture.
>
> Importantly, verifying Transformers is not just about dealing with different nonlinearities by directly using methods from previous work. It is significantly more challenging in the following aspects:
>
> 1. Cross-nonlinearity:  cross-nonlinearity appears in nonlinear functions involving two variables under perturbation. Deriving the linear relaxations for such functions is different and more challenging than that for univariate functions. The gradient descent based approach for RNN in (Ko et al., 2019) is inefficient (hundreds of iterations needed) and poses a computational challenge for the much more complex transformer verification problem. In contrast, we derive a novel closed-form linear relaxation in $O(1)$ complexity.
>
> 2. Cross-position dependency: this is special to self-attention, and there is no such dependency in feed-forward networks or RNNs verified by previous works. Resolving the cross-position dependency is critical for Transformers, otherwise the algorithm would be too slow. As the ablation study (Table 3 in revised paper) shows, if we directly extend the fully backward bounding framework (CROWN) to Transformers, it is much slower than our “backward & forward” algorithm which resolves the cross-position dependency. Our novel “backward & forward” algorithm reduces the time complexity by a factor of $O(n)$ ($n$ is the input length) compared to the fully backward algorithm, and still produces comparably tight bounds (much tighter than those by the fully forward algorithm).
>
> 3. Additionally, we have demonstrated an application beyond robustness verification using our techniques -- identifying important words (Table 4 in our revised paper). Thus, we believe that our proposed algorithm has enough novelty in both methodology and empirical study.
>
>
> We thank the reviewer again for the helpful comments, and we have addressed all the “weakness bullets” raised by the reviewer (response for point 3 in another reply below). We hope the reviewer can re-evaluate our paper based on our response.

---

### Official Review · AnonReviewer3 · 2019-10-23
**Official Blind Review #3**

**Rating:** 6

**Review:**

Summary:
This paper builds upon the CROWN framework (Zhang et al 2018) to provide robustness verification for transformers. The  CROWN framework is based upon the idea of propagating linear bounds and has been applied to architectures like MLP, CNNs and RNNs. However, in Transformers, the presence of cross-nonlinearities and cross-position dependencies makes the backward propagation of bounds in CROWN computationally intensive. A major contribution of this paper is to use forward propagation of bounds in self attention layers along with the usual back-propagation of bounds in all other layers. The proposed method provides overall reduction in computational complexity by a factor of O(n). Although the fully forward propagation leads to loose bounds, the mixed approach (forward-backward) presented in this work provides bounds which are as tight as fully backward method.


Strength:
Use of forward propagation to reduce computational complexity is non-trivial
Strong results on two text classification datasets:
Lower bounds obtained are significantly tighter than IBP
The proposed method is an order of magnitude faster than fully backward propagation, while still maintaining the bounds tight.

Weakness:
Experiments only cover the task of text classification. Experiments on other tasks utilizing transformers would have made the results stronger.
The paper makes the simplifying assumption that only a single position of an input sentence will be perturbed.  They claim that generalizing to multiple positions is easy in their setup but that is not supported.   The paper needs to declare this assumption early on in the paper (abstract and intro). As far as I could tell, even during the experiments they perturb a single word at a time.

The paper is more technical than insightful.  I am not at all convinced from a practitioners viewpoint that such bounds are useful.  However, given the hotness of this topic, someone or the other got to work out the details.   If the math is correct, then this paper can be it.

The presentation requires improvement.  Some parts, example, the Discussion section cannot be understood.


Questions:
The set-up in the paper assumes only one position in the input sequence is perturbed for simplicity. Does the analysis remain the same when multiple positions are perturbed?

Suggestions:
A diagram to describe the forward and backward process would significantly improve the understanding of the reader.

In Table 3, I am surprised that none of the sentiment bearing words were selected as the top-word by any of the methods.  Among the ‘best’ words,  the words chosen by their method does not seem better than those selected by the grad method.
Several typos in the paper: spelling mistakes in “obtain a safty guarantee”, poor sentence construction in “and independent on embedding distributions”, subject verb disagreement in “Upper bounds are discrete and rely on the distribution of words in the embedding space and thus it cannot well verify”.

I have not verified the math to see if they indeed compute a lower bound.


**Experience Assessment:**

I have read many papers in this area.

**Review Assessment: Checking Correctness Of Derivations And Theory:**

I assessed the sensibility of the derivations and theory.

**Review Assessment: Checking Correctness Of Experiments:**

I carefully checked the experiments.

**Review Assessment: Thoroughness In Paper Reading:**

I read the paper at least twice and used my best judgement in assessing the paper.

---

> ### Author Response · Authors · 2019-11-13
> **We have extended our method to multiple perturbed positions and provided additional experiments. We also fixed the table on identifying important words and added a diagram for the forward and backward processes.**
>
> We thank the reviewer for correctly identifying the main contributions of our paper and providing the very constructive feedback. We have revised our paper accordingly, and we provide answers below:
>
> (1) Regarding identified important words:
>
> We believe the presentation of the table caused a misunderstanding. In our initial submission, listed words were most important (or least important) words identified by different methods, and the red/bold words were *improperly* selected words by each method (less red words are better, and our method has the least red words). We have updated the table (Table 4 in our new revision) and captions for a better illustration, as detailed below.
>
> In our new table, boldfaced words are considered to have strong sentiment polarities. For the rows of “most important” words, baselines improperly select more words with no sentiment polarity (words that are not boldfaced in the revised table). E.g., for the gradient method, “dinner” and “food” are improperly selected, while our method chooses only one word (“level”) improperly. We also have quantitative results evaluated on more examples (column “Score (SST)” in the table) showing that the most important words identified by our method are indeed more important than those by baselines, and the least important words identified by our method are indeed less important.
>
> We hope it is now clear that our proposed method works better on identifying important words. You may also refer to our response to reviewer #1, who had the same confusion.
>
> (2) Regarding perturbing multiple positions:
>
> Our algorithm can be easily generalized to perturbations on multiple positions, and we have revised our methodology section for this generalization. We have also added experiments about perturbing two positions in Table 2 of the revised version, and they are consistent with previous conclusions. To support multi-position perturbation, we mainly need to modify the bound propagation to the input layer. We change from representing the bounds as linear functions of the one perturbed position ${\rm x}^{(r)}$ to linear functions of multiple perturbed positions ${\rm x}^{(r_1)}, {\rm x}^{(r_2)}, \cdots, {\rm x}^{(r_t)}$ ($t$ is the number of perturbed positions), and also accordingly modify the part of taking global bounds (Eq. (3), (4)) and reusing previous linear bounds (paragraph “backward process to self-attention layers”). Please refer to our revised paper for more details. However, the number of perturbed positions should still be small, since imperceptible adversarial examples may not have many perturbed positions (Gao et al., 2018; Ko et al., 2019).
>
> (3) Regarding the scope of tasks:
>
> As this is the first paper for verifying Transformers, we focus on developing an algorithmic framework for verification. Existing works on robustness verification also commonly focus on classification (Weng et al., 2018; Zhang et al., 2018; Ko et al., 2019), since the robustness of classification tasks has been well-defined. As long as the specification of robustness can be defined as a special case of Eq. (1), our proposed method can be applied to any model consisting of Transformer encoder/decoder in other complex NLP tasks, such as machine translation, question answering, parsing, or even tasks beyond languages. For the sake of simplicity and due to space limitation, we leave empirical analysis of these complex applications as future work, since in many cases there is even no well-accepted definition of robustness, unlike in classification tasks.
>
> (4) Regarding a diagram for the forward and backward processes:
>
> This is a great suggestion. We have provided a diagram in Appendix A to help readers understand the difference between forward, backward, and our novel backward&forward bound propagation processes.
>
> Lastly, thank you for pointing out the typos and we have fixed them. We improved writing of our paper in our revision, and we will continue working on polishing it.
>
>
>
> We greatly appreciate the helpful comments by the reviewer, and we hope the reviewer has a better understanding of our paper now. Feel free to let us know if there is anything still unclear, and we hope the reviewer can re-evaluate our paper based on our response.

---

### Official Review · AnonReviewer1 · 2019-10-26
**Official Blind Review #1**

**Rating:** 6

**Review:**

Contributions:
This paper develops an algorithm for verifying the robustness of transformers with self-attention layers when the inputs for one input word embedding are perturbed. Unlike previous work the present work can deal with cross nonlinearity and cross position dependency and the lower bounds derived in the paper are much tighter than the Interval Boundary Propagation (IBP) method which uses backward propagation. The core contribution is expounded by developing bounds for multiplication (xy) and division (x/y) and using this to compute tight bounds on self-attention layer computations. Further by introducing a forward process then combining it with a backward process, they can substantially reduce computation time as compared to the IBP method (in experiments demonstrated to be over an order of magnitude).

Experimental results: Very reasonable and well thought out experiments demonstrate that a) the lower bounds produced by the method are very tight (much better than IBP for instance), and the backward&forward approach is an order of magnitude faster than the fully backward approach though they are both equally tight (the fully forward method is much looser by contrast)

The experiments on using the bound to detect the word whose perturbation will cause biggest impact (ie the most important word for explaining the decision) is much less convincing. In particular in Table 3, the second example shows that the method identifiews "a", "the" "our" etc, which at least do not appear to be most salient to this reviewer. [Question to authors: perhaps I misunderstood something here -- if so please explain and help me understand it better].

Evaluation / Suggestions.
Overall I liked the paper quite a bit. I may not have fully understood something about table 3 and would welcome some explanation about why the results shown there indicate that their method works better int he second example.



**Experience Assessment:**

I do not know much about this area.

**Review Assessment: Checking Correctness Of Derivations And Theory:**

I assessed the sensibility of the derivations and theory.

**Review Assessment: Checking Correctness Of Experiments:**

I assessed the sensibility of the experiments.

**Review Assessment: Thoroughness In Paper Reading:**

I read the paper thoroughly.

---

> ### Author Response · Authors · 2019-11-13
> **We have improved the table on important words in our paper; sorry for the misleading presentation**
>
> We thank the reviewer for the encouraging feedback! We believe the presentation of this table caused a misunderstanding. We have improved Table 3 (which becomes Table 4 in our revised paper) for a better presentation of our results. Our proposed method outperforms other baselines.
>
> Previously, we presented two sets of results as indicated by the “type” in the first column. The first part (“most”) shows words that are the MOST salient while the second part (“least”) shows words that are the LEAST salient. Therefore, words such as “a”, “the”, “our” are indeed the LEAST important words identified by our algorithm as expected. We have updated the table and the captions to make them more clear: we have separated the list of “most important” words and “least important” words in different rows, and used boldfaced words to indicate words with strong sentiment polarities; boldfaced words should appear as “most important” words rather than “least important” words. Our proposed method clearly performs the best.
>
> We hope our response has resolved your question. Please let us know if you have any further questions regarding our paper, and we will appreciate it if you can consider increasing the rating.

---

### Decision · Program_Chairs · 2019-12-19

**Decision:**

Accept (Poster)

**Comment:**

A robustness verification method for transformers is presented. While robustness verification has previously been attempted for other types of neural networks, this is the first method for transformers.

Reviewers are generally happy with the work done, but there were complaints about not comparing with and citing previous work, and only analyzing a simple one-layer version of transformers. The authors convincingly respond to these complaints.

I think that the paper can be accepted, given that the reviewers' complaints have been addressed and the paper seems to be sufficiently novel and have practical importance for understanding transformers.